# WHAT SCALES IN CROSS-ENTROPY SCALING LAW?

**Junxi Yan, Zixi Wei, Qingyao Ai, Yiqun Liu, Jingtao Zhan**[*]
Tsinghua University
`yanjx21@gmail.com, zixiwei3@gmail.com, aiqy@tsinghua.edu.cn`
`yiqunliu@tsinghua.edu.cn, jingtaozhan@tsinghua.edu.cn`

## ABSTRACT

The cross-entropy scaling law has long served as a key tool for guiding the development of large language models. It shows that cross-entropy loss decreases in a predictable power-law rate as the model size increases. However, recent evidence indicates that this law breaks down at very large scales: the loss decreases more slowly than expected, which causes significant trouble for developing large language models. In this paper, we hypothesize that the root cause lies in the fact that cross-entropy itself does not truly scale; instead, only one of its hidden components does. To investigate this, we introduce a novel decomposition of cross-entropy into three parts: **Error-Entropy**, **Self-Alignment**, and **Confidence**. We show both theoretically and empirically that this decomposition precisely captures the training dynamics and optimization objectives. Through extensive experiments on multiple datasets and 32 models spanning five orders of magnitude in size, we find that only error-entropy follows a robust power-law scaling, while the other two terms remain largely invariant. Moreover, error-entropy constitutes the dominant share of cross-entropy in small models but diminishes in proportion as models grow larger. This explains why the cross-entropy scaling law appears accurate at small scales but fails at very large ones. Our findings establish the error-entropy scaling law as a more accurate description of model behavior. We believe it will have wide applications in the training, understanding, and future development of large language models.

## 1 INTRODUCTION

The cross-entropy scaling law has played a vital role in the development of large language models. This empirical law states that as model size and dataset size increase, the cross-entropy loss decreases in a predictable power-law manner (Kaplan et al., 2020). Its influence is both practical and theoretical. On the practical side, it has become an indispensable tool for training large language models, such as balancing model parameters and data scale (Hoffmann et al., 2022), extrapolating performance from smaller models to larger ones (Wei et al., 2022a), and tuning training hyperparameters (Kadra et al., 2023; Li et al., 2025). On the theoretical side, it has opened a gateway for understanding the principles of intelligence itself. Researchers have proposed various theories to explain why such a law emerges (Bahri et al., 2024; Zhang, 2024; Michaud et al., 2024), hoping that these explanations will shed light on the nature of artificial intelligence.

However, the cross-entropy scaling law has recently faced growing skepticism, both in practice and in theory. Practitioners have raised concerns about whether the law can continue to provide accurate predictions on larger scales. Empirical studies show that while cross-entropy decreases with an accurate power-law trend for small models, this trend becomes noticeably slower for very large models (Kaplan et al., 2020; Chen et al., 2025). On the theoretical side, the situation is equally unsettled. Although most existing theoretical frameworks can somehow prove the scaling law for error-based metrics, such as mean squared error (Lyu et al., 2025), they cannot directly generalize to cross-entropy loss. Thus, the theoretical foundation of this law is still unclear. These problems have fueled skepticism about whether cross-entropy truly scales, and in turn, they undermine confidence in scaling up models as a reliable path forward.

---

[*]Corresponding author

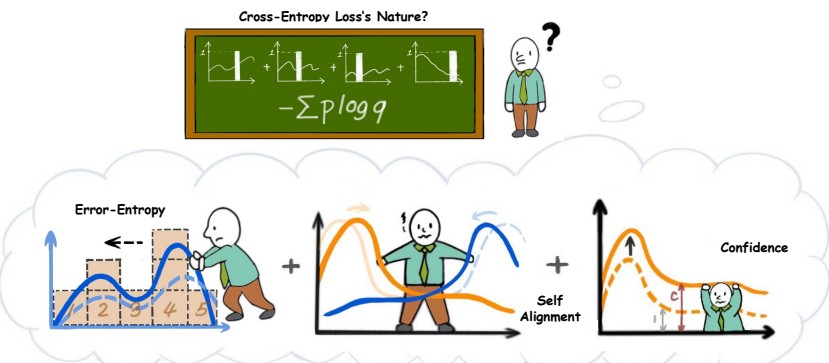

Figure 1: Decomposition of the Cross-Entropy Loss. The upper panel illustrates the cross-entropy loss. The lower panel presents the three components decomposed from cross-entropy. The blue curve denotes the Rank-based Error (RBE) distribution, and the orange curve represents the distribution of the ground-truth scores. Error-Entropy pushes the ground-truth tokens towards higher ranks. Self-Alignment aligns the probability score distribution with RBE distribution. And Confidence term increases the norm of the probability score.

In this work, we hypothesize that it is not cross-entropy loss itself that truly scales, but rather a dominant component hidden within it. This component gives the illusion that the cross-entropy itself follows a scaling law. Identifying this scaling component would therefore be of significant value. On the practical side, it would provide a more reliable law to guide the development of large language models. On the theoretical side, it could offer a better objective for investigating the principles of artificial intelligence. Motivated by this, the research question of this paper is:

*What scales in cross-entropy scaling law?*

To address this question, we begin by introducing a novel decomposition of cross-entropy. Specifically, we propose an error-based metric rooted in ranking, termed *Rank-based Error (RBE)*. Unlike cross-entropy, which measures the probability score of the correct token, RBE equals the rank of the correct token. For example, if four other tokens are scored higher than the ground truth, then the RBE is 4. Building on this notion, we decompose cross-entropy exactly into the sum of three terms: **Error-Entropy**, **Self-Alignment**, and **Confidence**, which is illustrated in Fig. 1. Among them, Error-Entropy measures the entropy of the RBE distribution. Minimizing this quantity effectively pushes the correct token toward higher ranks and thus makes the model's predictions more accurate. The other two terms characterize how the model aligns probability scores with the RBE distribution. Through both qualitative and quantitative analysis, we demonstrate that this decomposition accurately reflects the training dynamics of language models.

Building on this decomposition, we investigate the scaling behavior of these components and find that only Error-Entropy truly scales. We refer to it as the Error-Entropy Scaling Law. Concretely, we conduct experiments on multiple real-world language datasets using 32 models spanning five orders of magnitude in size. [1] We then evaluate the scaling behavior of cross-entropy alongside its three components. The results are clear. (1) Error-Entropy decreases according to a power law with model size, whereas the other two terms remain random for different model sizes. (2) The power-law fit of Error-Entropy is even better than that of cross-entropy. It indicates that the Error-Entropy scaling law is potentially the reason why cross-entropy approximately exhibits scaling behavior.

We believe that our decomposition of cross-entropy and the discovery of the Error-Entropy scaling law provide a better description of how language models actually scale, and open the door to broad applications. As one example, this framework helps resolve a long-standing puzzle (Kaplan et al., 2020; Chen et al., 2025): why does the cross-entropy scaling law appear accurate for small models but break down for larger ones? The answer lies in the proportion of Error-Entropy. It dominates in small models, making cross-entropy follow a clean power law, but its declining share in larger models allows non-scaling terms to dominate, breaking the law. This specific example highlights how Error-Entropy can sharpen our understanding of model behavior. More broadly, we expect that

---

[1]The code is available at `https://github.com/yanjx2021/RethinkCE`.

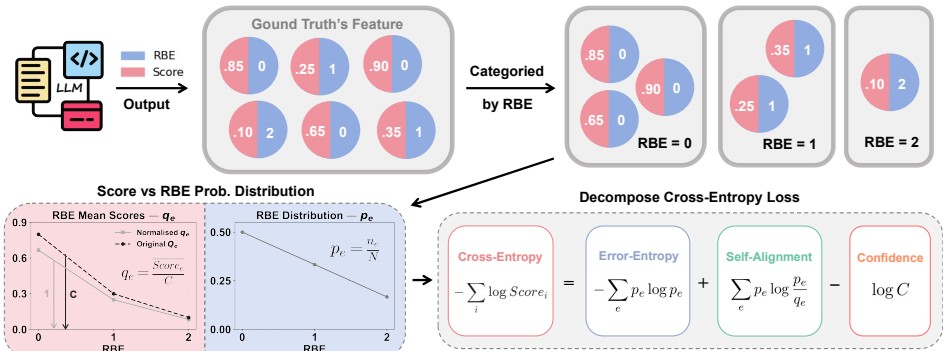

Figure 2: Overview of the decomposition. Rank-based Error (RBE) is the ranking of the ground-truth token. Model's predictions are grouped based on RBE values. For each group, we compute its proportion $p_e$ (termed RBE distribution), the normalized prediction score $q_e$, and the norm of scores $C$. Based on these definitions, cross-entropy can be mathematically decomposed into three components: Error-Entropy, Self-Alignment, and Confidence.

it will have wide-ranging implications, such as guiding the design and training of large language models, probing model mechanisms, and finding fundamental theories of artificial intelligence.

## 2 RELATED WORK

**Neural Scaling Laws.** Early empirical studies had already suggested that deep learning performance often follows predictable scaling behavior across tasks and modalities (Hestness et al., 2017). Kaplan et al. (2020) first systematically observed a power-law relationship between model performance, parameter count, dataset size, and compute in language modeling, and proposed compute allocation rules; Hoffmann et al. (2022) later refined this into a compute-optimal scaling strategy. Henighan et al. (2020) extended the cross-entropy scaling law to images, videos, multimodal tasks, and mathematics, while Zhai et al. (2022) and Ghorbani et al. (2021) validated similar behaviors in vision and machine translation. Beyond these, Hernandez et al. (2021); Barnett (2024) studied scaling in transfer learning. Although numerous works have documented empirical scaling laws, the *fundamental question* of why cross-entropy decreases according to a power-law remains unclear. Some studies have attempted to explain scaling behavior for error-based metrics such as mean squared error (Lyu et al., 2025), but these results are difficult to generalize to cross-entropy, the standard loss for training classifiers and language models. In this work, we address this gap by identifying the components of cross-entropy that truly drive its scaling behavior.

**Cross-Entropy Loss.** In the neural network domain, prior studies about cross-entropy have mainly examined its properties regarding consistency, robustness, calibration, and regularization (Guo et al., 2017; Fort & Scherlis, 2018; Wei et al., 2022b; Soudry et al., 2024; Vysogorets et al., 2024). For instance, Mao et al. (2023) conducted a systematic theoretical analysis of cross-entropy and its variants, deriving H-consistency bounds. However, these analyses do not investigate whether—and how—such intrinsic properties of cross-entropy interact with macroscopic scaling behavior. Our work complements these directions by decomposing cross-entropy into components with clearer operational meaning and studying how these components scale with model size.

## 3 FROM CROSS-ENTROPY TO ERROR-ENTROPY

In this section, we decompose cross-entropy into three components and show how these components relate to model performance. First, Sec. 3.1 introduces a new metric. Second, Sec. 3.2 uses this metric to mathematically decompose cross-entropy loss and then empirically examines the correctness of this decomposition. Finally, Sec. 3.3 compares the three components and shows that one of the components, termed Error-Entropy, is closely tied to model performance.

## 3.1 Rank-based Error

Which better reflects the performance of a language model, the probability assigned to the correct token, or its rank among all tokens? While cross-entropy is defined in terms of probabilities, we argue that rank provides a more robust indicator. Probabilities are easily manipulated: during inference, techniques such as temperature scaling, top-k sampling, and nucleus (top-p) sampling directly alter probability values, and such changes also affect the cross-entropy loss. By contrast, the relative ordering of tokens is much harder to distort. None of the common sampling strategies changes the ordering. For this reason, we view the rank position of the correct token as a more fundamental measure of model performance than its raw probability scores.

Therefore, we propose a new metric, *Rank-based Error (RBE)*, which equals the rank of the ground-truth token predicted by the language model. It serves as a measure of how far the prediction deviates from the right answer. Formally, let $i$ be the index of the input data, and $v_i$ be the correct token for this data. $\mathcal{V}$ is the vocabulary. We use $s_{v_i}$ to denote the model's output probability score for $v_i$. Thus, *RBE* for $v_i$ is defined as:

$$\text{RBE}(v_i) = \sum_{v \in \mathcal{V}} \mathbf{1}\{s_v > s_{v_i}\}, \tag{1}$$

where $\mathbf{1}\{\cdot\}$ denotes the indicator function that equals 1 if the condition is true and 0 otherwise. Intuitively, a smaller RBE indicates that the model ranks the correct token near the top and thus corresponds to better model performance.

Based on RBE, we define two distributions. First, we treat RBE as a random variable and analyze its probability distribution over the corpus. Second, we group model predictions by RBE and compute the mean score for each group. An illustration is shown in Fig. 2. The details are as follows:

**RBE Distribution.** We define the *RBE distribution* $p_e$ as the probability mass function of RBE:

$$p_e = \Pr\left[\text{RBE}(v_i) = e \mid v_i \in \mathcal{D}\right], \tag{2}$$

where $\mathcal{D}$ denotes the collection of all output ground-truth tokens in the test corpus. Intuitively, $p_e$ characterizes how often the ground-truth token appears at rank $e$ in model predictions. For a well-trained model, we expect $p_e$ to concentrate more heavily on smaller ranks $e$, indicating that the model is more likely to place the ground-truth token near the top of its prediction list.

**Score Distribution.** Complementary to $p_e$, we define the *Score distribution* $q_e$ by grouping the predictions with the same RBE values and computing their average scores. Formally, $q_e$ is the geometric mean of $\{s_{v_i}\}$ for those $v_i$ with $\text{RBE}(v_i) = e$:

$$Q_e = \text{GeoMean}(\{s_{v_i} \mid \text{RBE}(v_i) = e\}), \tag{3}$$

where the geometric mean is $\text{GeoMean}(\{a_j\}_{j=1}^m) = \left(\prod a_j\right)^{1/m}$. We further normalize $\{Q_e\}$ as:

$$q_e = \frac{Q_e}{C}, \quad C = \sum_e Q_e. \tag{4}$$

$q_e$ is the normalized score distribution, and $C$ is the norm of the prediction scores. A larger $C$ indicates that the model is generally more confident in its predictions.

Next, we will use $p_e$, $q_e$, and $C$ to decompose cross-entropy.

## 3.2 Mathematical Decomposition

This section mathematically decomposes cross-entropy. We start with its definition: cross-entropy measures the divergence between the predicted distribution and the ground-truth distribution. In language modeling, the ground-truth distribution is usually implemented as a one-hot vector that takes the value 1 at the ground-truth token and 0 elsewhere. Let $s_{v_i}$ denote the score of ground-truth token $v_i$. Then cross-entropy loss is $-\log s_{v_i}$. Given a corpus, it is the average of all $-\log s_{v_i}$.

The core of our decomposition is to group cross-entropy terms by their associated *RBE* values. Consider a corpus with $N$ inputs. For each input, the model produces a cross-entropy item. We

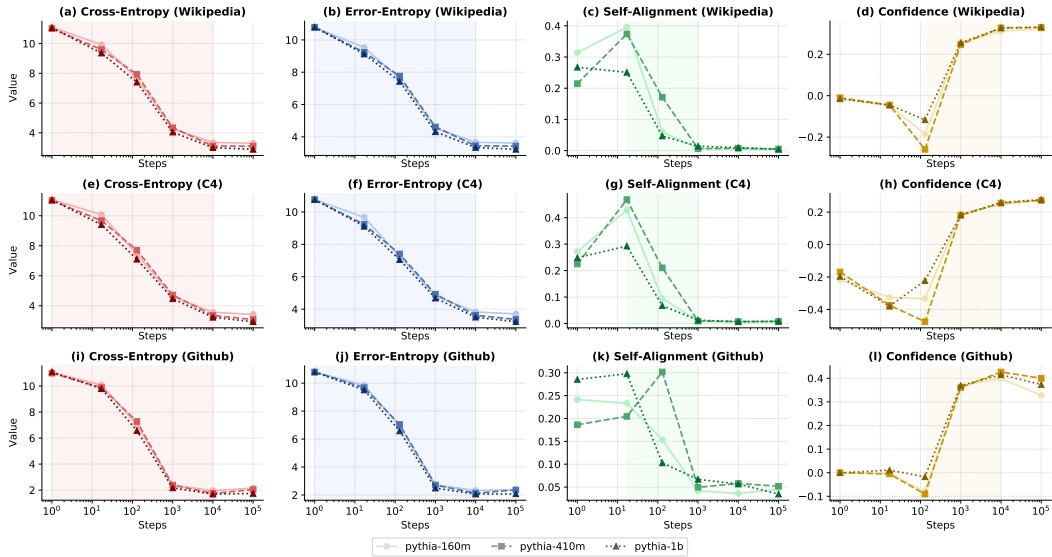

Figure 3: Evolution of cross-entropy and its decomposed components during training on three datasets. All components are indeed optimized during training, as suggested by the mathematical deduction: Error-Entropy steadily decreases, Self-Alignment declines in the end, and Confidence term increases. The shaded regions highlight the training progress with the most rapid metric changes.

group these cross-entropy items by putting those with the same RBE values in the same group. Formally, the cross-entropy loss can be rewritten as:

$$\mathcal{L}_{\mathrm{CE}} = -\frac{1}{N}\sum_{i=1}^{N}\log s_{v_i} = -\frac{1}{N}\sum_{e}\left(\sum_{i:\,\mathrm{RBE}(v_i)=e}\log s_{v_i}\right) \tag{5}$$

Let $n_e$ be the number of items in the group whose RBE value equals $e$. By converting the summation of logarithms into a logarithm of products, the above expression can be further rewritten as:

$$\mathcal{L}_{\mathrm{CE}} = -\sum_{e}\underbrace{\frac{n_e}{N}}_{p_e}\cdot\frac{1}{n_e}\log\left(\prod_{i:\,\mathrm{RBE}(v_i)=e} s_{v_i}\right) = -\sum_{e} p_e\cdot\log\underbrace{\left(\prod_{i:\,\mathrm{RBE}(v_i)=e} s_{v_i}\right)^{\frac{1}{n_e}}}_{Q_e}. \tag{6}$$

Note that the above equation uses Eq. 2 and Eq. 3 to replace the variables with $p_e$ and $Q_e$. Next, according to Eq. 4, each $Q_e$ can be further expressed as $Q_e = C \cdot q_e$, yielding:

$$\mathcal{L}_{\mathrm{CE}} = -\sum_{e} p_e\log Q_e = -\sum_{e} p_e\big(\log q_e + \log C\big) = -\sum_{e} p_e\log q_e - \log C\cdot\sum_{e} p_e. \tag{7}$$

Since $p_e$ is a probability distribution, we can simplify $\sum_e p_e = 1$. Moreover, the cross-entropy between $p_e$ and $q_e$ can be rewritten as the sum of the Shannon entropy of $p_e$ and the KL divergence between $p_e$ and $q_e$, yielding:

$$\mathcal{L}_{\mathrm{CE}} = -\sum_{e} p_e\log\left(p_e\cdot\frac{q_e}{p_e}\right) - \log C = \underbrace{-\sum_{e} p_e\log p_e}_{\text{Error-Entropy}} + \underbrace{\sum_{e} p_e\log\frac{p_e}{q_e}}_{\text{Self-Alignment}} - \underbrace{\log C}_{\text{Confidence}}. \tag{8}$$

Based on the above mathematical derivation, we decompose the cross-entropy loss into three components: (i) Error-Entropy: the Shannon entropy of $p_e$, (ii) Self-Alignment: the KL divergence between $p_e$ and $q_e$, and (iii) Confidence: the logarithm of the normalization constant $C$.

We analyze the training process to empirically verify that these components constitute the cross-entropy loss and are optimized throughout training.[2] Fig. 3 shows their values during the entire

---

[2]Experimental details for this section are provided in App. A.1.

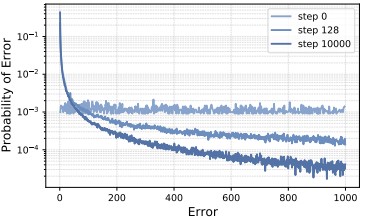
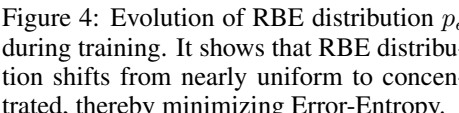
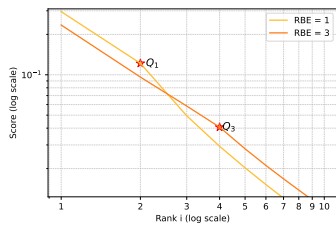

Figure 4: Evolution of RBE distribution $p_e$ during training. It shows that RBE distribution shifts from nearly uniform to concentrated, thereby minimizing Error-Entropy.

Figure 5: Output score distribution when ground-truth tokens are ranked at 2 and 4, respectively. Scores drop for tokens ranked lower than ground-truth tokens, thereby increasing Confidence term.

training process. We can see that all three components are optimized during training. *Error-Entropy* and *Self-Alignment* decreases, and *Confidence* term increases. Moreover, the different magnitudes of the three components lead to a difference in the order of optimization during training. Since *Error-Entropy* has the largest initial value, the model focuses on reducing it first in the early stages of training. In contrast, *Self-Alignment* and *Confidence* terms are much smaller in magnitude, and the model begins to optimize them only after *Error-Entropy* has been largely minimized. Therefore, our decomposition captures the dynamics of training in a clear and interpretable way.

### 3.3 COMPARISON OF THE THREE COMPONENTS

Now, we compare the three components based on their optimization objectives.

**Error-Entropy:** Error-Entropy uses the form of Shannon entropy to quantify how severely the model makes mistakes. Mathematically, minimizing this term requires the RBE distribution to become as concentrated as possible. Achieving this means the model shall learn to identify which token is the correct one. Fig. 4 illustrates the evolution of the RBE distribution during training. At step 0, when the model is untrained, the RBE distribution is nearly uniform. This indicates that the model has no understanding of which token is correct and is essentially ranking tokens at random. At this stage, Error-Entropy is maximal. As training progresses, optimization drives the RBE distribution to concentrate toward the head. This means that the model is increasingly able to recognize the correct token and place it at the top of the ranking. Therefore, optimizing error-entropy directly encourages the model to develop a clear distinction between right and wrong.

**Self-Alignment:** Self-Alignment uses KL divergence to measure the difference between the RBE distribution $p_e$ and the normalized score distribution $q_e$. Mathematically, minimizing self-alignment requires $q_e$ to exactly match $p_e$. This provides a new perspective on how to interpret a model's output scores. Unlike cross-entropy, which assumes that the predicted probabilities approximate the true distribution of language, self-alignment suggests that the model instead assigns probabilities based on its own likelihood of making errors. To illustrate this, Fig. 6 shows $p_e$ and $q_e$ distributions at different stages of training. Early in training, the two distributions diverge significantly. However, after sufficient training, they become almost indistinguishable. This indicates that the model learns to align its output probabilities with its internal error distribution. Since different models have different error distributions, they also produce different probability scores. If, by contrast, probabilities truly represented the single real language distribution as suggested by cross-entropy, we would not expect variation in probability scores across models, which deviates from empirical observations (Guo et al., 2017; Liang et al., 2023).

**Confidence:** Confidence term appears with a negative sign in our decomposition, meaning that it is increased during optimization. It is the magnitude of the probability scores of correct tokens, formally expressed as $\log \sum_e Q_e$. Note that $Q_e$ is the probability score for all tokens ranked at $e+1$ position. Thus, the maximum value of $Q_e$ is $1/(e+1)$, which corresponds to the case where the top $e+1$ tokens share equal probability scores while the rest tokens receive probability zero. If this maximum is achieved for every $e$, the Confidence term becomes $\log(\sum_i 1/i)$. Since the harmonic series diverges, the confidence term could be very large. Intuitively, this extreme case represents a highly confident model: it is confident that the tokens ranked lower than $e+1$ can never be correct answers and thus are assigned zero probability. This motivates the name Confidence term. Fig. 5

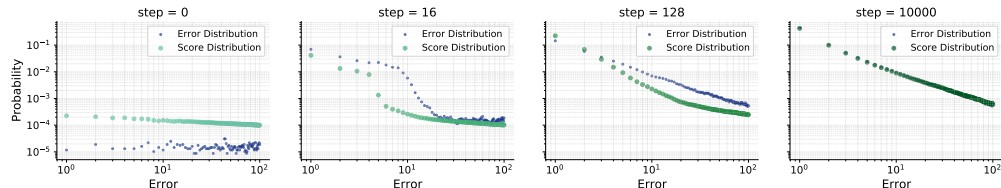

Figure 6: Evolution of RBE distribution $p_e$ and normalized score distribution $q_e$ during training. The two distributions gradually become identical, thereby minimizing the Self-Alignment term.

Table 1: Comparison of power-law fit quality for cross-entropy and its three components. Best values within each family are bolded. Error-Entropy achieves the highest $R^2$ in nearly all cases and even outperforms cross-entropy.

| $R^2$ | Wikipedia | | | | C4 | | | | GitHub | | | |
|---|---|---|---|---|---|---|---|---|---|---|---|---|
| | CE | EE | SA | CONF | CE | EE | SA | CONF | CE | EE | SA | CONF |
| Qwen | 0.9731 | **0.9753** | 0.9441 | 0.2977 | 0.9493 | **0.9529** | 0.8492 | 0.4840 | 0.9882 | **0.9896** | 0.9455 | 0.1371 |
| Pythia | 0.9448 | **0.9767** | 0.0190 | 0.812 | **0.9930** | 0.9825 | 0.3082 | 0.8366 | 0.7058 | **0.7328** | 0.0760 | 0.6965 |
| GPT2 | 0.9577 | **0.9734** | 0.7212 | 0.6518 | **0.9892** | 0.9872 | 0.3357 | 0.9444 | 0.0717 | **0.9262** | 0.0586 | 0.1167 |
| All | 0.8421 | **0.8626** | 0.3553 | 0.0202 | 0.8699 | **0.9012** | 0.2188 | 0.0492 | 0.6743 | **0.7229** | 0.3233 | 0.0203 |

illustrates how models behave in practice. It shows the output score distribution for two cases. In the first case, the correct token is ranked at 2 (RBE = 1). We can see that the probability mass drops sharply after the third token. In the second case, the correct token is ranked at 4 (RBE = 1). We can see that the drop occurs after the fifth token. Both cases echo our above analysis that models try to be confident in their predictions and assign small scores to lower-ranked tokens.

We believe that error-entropy best captures the performance of a language model. Unlike the other terms, error-entropy depends only on the ranking of tokens rather than their raw probability scores. Minimizing it requires the model to place the correct token ahead of incorrect ones, which directly reflects the ability to distinguish right from wrong. Moreover, because it is independent of probability values, error-entropy is immune to post-processing techniques such as temperature rescaling or top-p sampling, making it a robust metric. By contrast, both the self-alignment and confidence terms are tied to probability scores. While they describe how the model assigns scores to tokens, they do not fully capture model accuracy and are more vulnerable to distortions introduced by sampling strategies. In summary, through this decomposition, we obtain a fine-grained view of model behavior and identify error-entropy as the component that most faithfully reflects model performance.

## 4 ERROR-ENTROPY SCALING LAW

In this section, we systematically examine the *scaling behavior* of the decomposed components of the cross-entropy loss. Sec. 4.1 visualizes how different components change when model size is scaled up. Then Sec. 4.2 quantitatively examines the scaling law.

### 4.1 QUALITATIVE ANALYSIS

In this section, we study how the three decomposed components—*Error-Entropy (EE)*, *Self-Alignment (SA)*, and *Confidence (Conf)*, scale with model size.[3] The results are shown in Fig. 7.

We can see that, among the decomposed components, only *Error-Entropy* exhibits a clear scaling property. Specifically, in a log–log plot, we observe that *Error-Entropy* decreases approximately linearly with increasing model size, closely resembling that of the cross-entropy. In contrast, *Self-Alignment* does not improve with larger models but instead shows an overall upward trend, while *Confidence term* displays larger variance and lacks a consistent pattern.

Moreover, we find that *Error-Entropy* generally scales better than cross-entropy. In particular, most points for *Error-Entropy* lie closer to the fitted line. For example, on GitHub dataset, some data points of cross-entropy are far from the fitted line, while the points of error-entropy are generally closer. This suggests that *Error-Entropy* shall be the truly scaled part of cross-entropy.

---

[3]Experimental details for this section are provided in App. A.2.

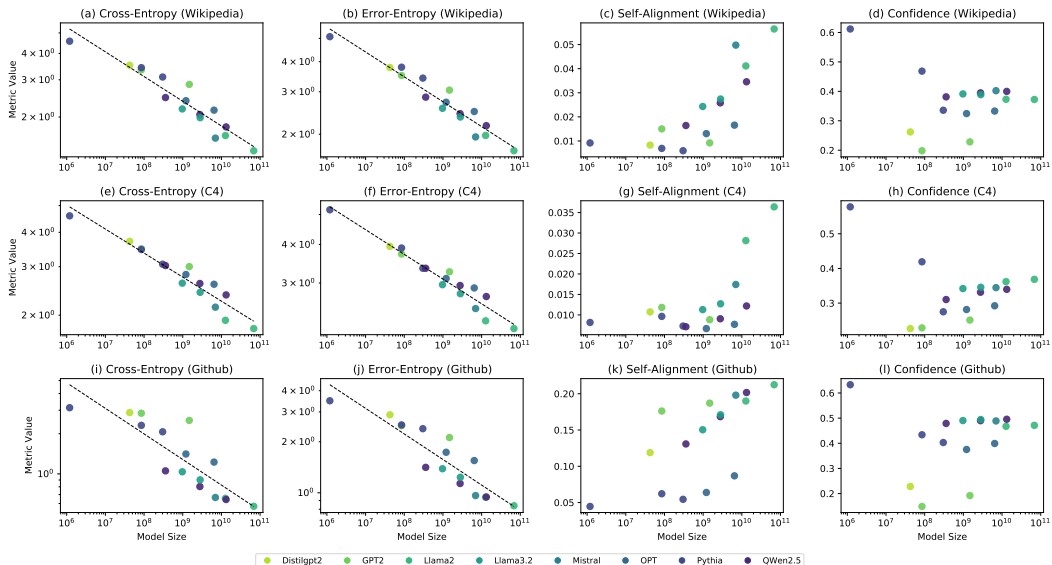

Figure 7: Illustration of cross-entropy and its decomposed components for different model sizes. Different rows correspond to different datasets. Only cross-entropy and Error-Entropy exhibit clear scaling behavior, with Error-Entropy lying closer to the fitted power-law line.

Table 2: Comparison of exponent difference ($|\Delta|$) for cross-entropy components. $|\Delta|$ measures the difference in scaling exponent between each component and cross-entropy. Best values are bolded. Results show that the scaling exponent of Error-Entropy is closest to cross-entropy.

| $|\Delta|$ | | Wikipedia | | | C4 | | | GitHub | |
|---|---|---|---|---|---|---|---|---|---|
| | EE | SA | CONF | EE | SA | CONF | EE | SA | CONF |
| Qwen | **0.0104** | 0.2347 | 0.0786 | **0.0069** | 0.1599 | 0.0652 | **0.0214** | 0.2138 | 0.1234 |
| Pythia | **0.0057** | 0.0494 | 0.0059 | **0.0038** | 0.0969 | 0.0354 | **0.0178** | 0.1171 | 0.0344 |
| GPT2 | **0.0069** | 0.1585 | 0.1076 | **0.0059** | 0.0904 | 0.0883 | **0.0352** | 0.2126 | 0.0866 |
| All | **0.0147** | 0.2678 | 0.1002 | **0.0058** | 0.1672 | 0.0625 | **0.0406** | 0.3307 | 0.2116 |

## 4.2 QUANTITATIVE ANALYSIS

We further examine the scaling properties of the components with quantitative fitting experiments.

We use two metrics, namely $R^2$ to examine how well the results follow a power-law rate, and $\Delta$ to examine how close the scaling behavior is to that of cross-entropy. They are defined as follows. Let $N$ be the number of model *non-embedding* parameters. For each metric $M \in \{\text{CE}, \text{EE}, \text{SA}, \text{Conf}\}$, we perform a log-log linear regression across models and report the goodness of fit $R^2$:

$$\log |M| = c_M + \alpha_M \log N \tag{9}$$

where $c_M$ is the intercept and $\alpha_M$ is the scaling exponent (slope). We further define the *scaling difference* between each decomposed component and cross-entropy to quantify their scaling consistency: $|\Delta_M| = |\alpha_M - \alpha_{\text{CE}}|$, where smaller $|\Delta_M|$ indicates closer alignment with cross-entropy. The results across datasets are summarized in Table 1 (for $R^2$ values) and Table 2 (for $\Delta$ values).

The quantitative experiments further support our hypothesis that only *Error-Entropy* consistently exhibits scaling behavior. Specifically: (i) *Error-Entropy* scales robustly across both single-model families (Qwen, Pythia, GPT-2) and the mixed-model setting, achieving strong fits with $R^2$ values close to 0.9 across Wikipedia, C4, and GitHub. (ii) *Self-Alignment* lacks a clear power-law pattern. Apart from the Qwen series showing relatively strong fitting, most other cases yield lower $R^2$ values. Moreover, it displays the largest $|\Delta|$ among the components, indicating the greatest deviation from cross-entropy. (iii) *Confidence* term is effectively signal-poor. Its $R^2$ values are highly unstable across model families and datasets. In the mixed-model case, they drop to extremely low levels (0.1002, 0.0625, 0.2116), suggesting the absence of consistent scaling. Although its $|\Delta|$ is not

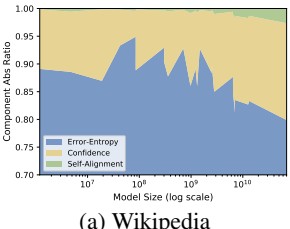
(a) Wikipedia

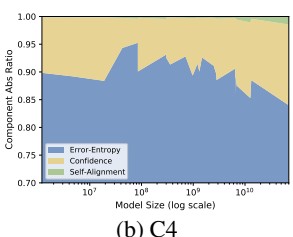
(b) C4

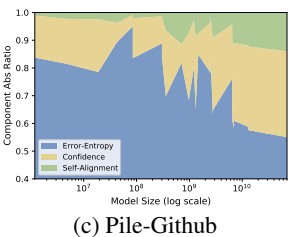
(c) Pile-Github

Figure 8: Relative ratio of decomposed components at different model sizes. For small models, Error-Entropy dominates (about 80–90%). Its share gradually decreases as model size grows.

always large, the overall evidence implies that *Confidence* term primarily reflects noise-dominated curves rather than a stable power-law relationship.

Taken together, these results demonstrate that *Error-Entropy* is the true driver of cross-entropy scaling. Its $R^2$ values exceed those of cross-entropy in nearly all settings, which suggests that it is the truly scaled part within cross-entropy. The observation that *Error-Entropy* consistently achieves the smallest $|\Delta|$ across all cases shall be the result of *Error-Entropy* being the true scaling driver of cross-entropy. Therefore, these results indicate that cross-entropy scaling originates in the behavior of *Error-Entropy* Scaling Law.

## 5 APPLICATION OF ERROR-ENTROPY SCALING LAW

Our decomposed result and the discovered Error-Entropy scaling law better characterize the model scaling behavior. We believe they will have wide applications for LLMs. In this section, we provide an example in using them to understand the puzzle of slowing cross-entropy scaling law.

The puzzle is that cross-entropy scales precisely at a power-law rate for small models, and yet slows down for very large models. Concretely, for small models, Kaplan et al. (2020) described the relationship between cross-entropy $\mathcal{L}_{\mathrm{CE}}$ and model scale $M$ as $\mathcal{L}_{\mathrm{CE}} \propto M^{-\alpha}$, which is a pure power-law form. Yet, when model size grows, OpenAI et al. (2024) found cross-entropy loss starts to slow down. To account for this problem, the expression is rewritten as $\mathcal{L}_{\mathrm{CE}} \propto M^{-\alpha} + bias$. Nowadays, researchers even start to find out cross-entropy slows down even more (Qin et al., 2025; Lourie et al., 2025). This phenomenon has been widely observed but poorly understood.

Our decomposition and Error-Entropy scaling law offer a natural explanation. Since only *Error-Entropy* scales reliably, the validity of cross-entropy scaling depends on how much of the total loss is contributed by Error-Entropy. Fig. 8 shows the relative ratio of the three components[4]. For small models, Error-Entropy dominates, accounting for nearly 90% of cross-entropy, and the overall loss therefore appears to follow a clean power law. For larger models, however, the proportion of Error-Entropy declines, while the non-scaling components—Self-Alignment and Confidence—take up a larger share, causing cross-entropy to deviate from the expected power-law. This explains the observed breakdown of cross-entropy scaling.

## 6 DISSCUSION AND FUTURE WORK

In this section, we further discuss the contributions and theoretical implications of our work. We analyze the deeper principles behind Error-Entropy in Sec. 6.1. We also outline possible training objectives and future directions motivated by this perspective in Sec. 6.2.

### 6.1 DEEPER PRINCIPLES BEHIND ERROR-ENTROPY

Although our definition of Error-Entropy (EE) arises from cross-entropy decomposition, we believe it reflects deeper underlying principles. In information-theoretic learning (ITL) and signal processing field, error entropy has been studied extensively (Erdogmus & Principe, 2002; Liu et al., 2006; Li et al., 2022). Our work establishes a link between these fields and large language models, offering a new perspective on model behavior and training dynamics.

---

[4]Experimental details are elaborated in App. A.2.2

While our formulation of Error-Entropy differs from the classic ITL version, the two are closely connected. ITL studies focus on nonparametric regression, where both input and output are real-valued. The error is defined as $e = y - \hat{y_{pred}}$, which makes optimizing the error distribution a natural objective. A common ITL-style error entropy is: $H(e) = -\int p(e) \log p(e)\, de$. In contrast, language modeling is a classification task where each token in the vocabulary is assigned a probability, and the goal is to raise the rank of the correct token. We define a rank-based error that restores an error distribution in this setting, and use it to decompose cross-entropy into error-related and probability-related components. Our error-entropy shares a similar form with the ITL definition, which links the probability-based formulation used in LLM training to the error-distribution perspective from ITL.

This connection paves the way for a new line of research. From an optimization perspective, techniques developed in ITL kernel-based methods for minimizing error entropy (Chen et al., 2013; 2019) may inspire new training objectives for language models. These approaches often use stochastic gradient methods in kernel spaces, and have been shown to handle heavy-tailed and non-Gaussian noise effectively. From a theoretical perspective, existing insights from ITL can be revisited in the LLMs setting to explain the observed scaling behavior of EE. In particular, the robustness of error-entropy under noisy or mismatched supervision, one of its original motivations in ITL (Tang & Li, 2011), may also help explain why EE scales more consistently than CE in large-scale models.

## 6.2 TRAINING OBJECTIVES INSPIRED BY ERROR-ENTROPY

In this subsection, we show that it is feasible to design practical training objectives inspired by Error-Entropy. EE itself is non-differentiable with respect to the logits, so we instead consider surrogate training objectives. Specifically, we define a loss that adds a penalty term on Confidence:

$$\mathcal{L}_\lambda = \text{CE} + \lambda \cdot \text{CONF}, \quad 0 < \lambda < 1. \tag{10}$$

We penalize the Confidence term because it is not directly related to ranking ability. As discussed in Section 3.3, increasing Confidence alone does not bring significant improvement in performance. Moreover, our empirical results (Fig. 8) shows that its share in cross-entropy grows with model size, which suggests that current training over-emphasizes this component. By reducing its weight in the loss, we aim to shift optimization toward EE, which is the component with more desired scaling.

Mathematically, the compensated loss is differentiable with respect to the logits, so it can be optimized with standard gradient-based training. In our notation, cross-entropy can be written as $\text{CE} = -\frac{1}{N}\sum_i \log s_i$, where $s_i$ is the score of the correct token for example $i$. The Confidence term takes the form $\text{CONF} = \log \sum_e Q_e$, where $Q_e$ is the mean score over examples with error level $e$. Differentiating $\mathcal{L}_\lambda$ with respect to $s_i$ yields (See App. D for the full derivation.)

$$\frac{\partial \mathcal{L}_\lambda}{\partial s_i} = -\frac{1}{N s_i}\left(1 - \lambda \frac{q_e}{p_e}\right), \tag{11}$$

where $p_e$ is the error distribution and $q_e$ is the score distribution. With original cross-entropy loss, once a token attains the correct rank, further increasing its probability only raises $q_e$ without improving $p_e$. The factor $\left(1 - \lambda \frac{q_e}{p_e}\right)$ encourages the model to keep $q_e$ closer to $p_e$, so that optimization focuses more on improving the error distribution captured by error-entropy.

Beyond such differentiable surrogates, error-entropy can also be used directly as a non-differentiable objective in reinforcement-learning style fine-tuning, where it serves as a reward signal that depends only on ranks. This avoids differentiating error-entropy directly and may provide a complementary route to align training with error-entropy.

## 7 CONCLUSION

In this paper, we propose a mathematical decomposition of the cross-entropy loss and conduct wide-ranged experiments to find out the truly scaled component. Results show that our decomposition clearly characterizes model behavior during training. Based on our decomposition, we find that *Error-Entropy* is the truly scaled part hidden within cross-entropy, while the other two components do not scale. The proposed decomposition and new scaling law provide a novel perspective to understand model behavior, such as how models assign probability scores and why the rate of cross-entropy scaling slows down. We believe they will facilitate a more fundamental understanding of model mechanics and may enable new training paradigms in the future.

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

## A  MORE DETAILS OF EXPERIMENTAL SETUP

### A.1  EXPERIMENTAL SETUP FOR TRAINING DYNAMICS

This section summarizes the general experiment settings used in our training-dynamics analyses. The detailed settings for each individual experiment are placed in Table 4.

**Models.**  We adopt the **Pythia** family (Biderman et al., 2023) at 160 M, 410 M, and 1 B parameters. Pythia is widely used in scaling-law and training-dynamics research because it (i) provides a consistent architecture across sizes, and (ii) publicly releases dense *training checkpoints*, enabling fine-grained temporal analysis without re-training. Unless otherwise noted, we evaluate checkpoints $\{0, 16, 128, 1\,000, 10\,000, 100\,000\}$, chosen to span *orders of magnitude* in optimization steps.

Table 3: Total number of evaluation tokens used for the main experiments.

|  | Wikipedia | C4 | Pile-GitHub |
|---|---|---|---|
| # Tokens | 704,185 | 417,140 | 1,091,481 |

**Datasets.** We use three English corpora: **Wikipedia** (20231101 snapshot) (Guo et al., 2020), which offers clean encyclopedic prose and is commonly included in pretraining pipelines; **C4** (Dodge et al., 2021), a large-scale web-derived corpus covering diverse styles and domains; and the **GitHub** subset of **The Pile** (Gao et al., 2020), which introduces code-like sequences and thus serves as a deliberately *harder* distribution to stress-test our decomposition.

**Sampling protocol.** Unless otherwise noted, we evaluate on the first *1 000* instances from each dataset (after shuffling with a fixed seed of 42). Note that these instances are full sequences containing many tokens; the total number of evaluation tokens is summarized in Table 3. We further verified robustness by repeating selected experiments with substantially larger sample sizes (e.g., 10k instances), and observed identical trends, indicating that our conclusions are insensitive to the particular sample count or ordering.

**Metrics.** For every evaluation token, we record the softmax score of the ground-truth token and its *Rank-based Error* $e$ (the rank position of the ground-truth token). From these, at each checkpoint we estimate the empirical Error distribution $p_e$, the Score distribution $q_e$, and the global Confidence terms statistic $C$. Then at the corpus level we report CE and its three components—*Error-Entropy (EE)*, *Self-Alignment (SA)*, and *Confidence (Conf)*—as defined in Sec. 3.2.

Table 4: Experimental setups for Fig. 3–5.

| Fig. | Data | Model & Checkpoints |
|---|---|---|
| 3 | Wikipedia (1k), C4 (1k), Pile-GitHub (1k) | Pythia-160M/-410M/-1B @ {0,16,128,1k,10k,100k} |
| 4 | Wikipedia (1k) | Pythia-160M @ {0,128,10k} |
| 5 | Wikipedia (1k) | Pythia-70M @ {32k} |
| 6 | Wikipedia (1k) | Pythia-410M @ {0,16,128,10k} |

## A.2 EXPERIMENTAL SETUP FOR SCALING

**Datasets.** We use three English corpora: **Wikipedia** (20231101 snapshot) (Guo et al., 2020), which offers clean encyclopedic prose and is commonly included in pretraining pipelines; **C4** (Dodge et al., 2021), a large-scale web-derived corpus covering diverse styles and domains; and the **GitHub** subset of **The Pile** (Gao et al., 2020), which introduces code-like sequences and thus serves as a deliberately *harder* distribution to stress-test our decomposition.

**Sampling protocol.** Unless otherwise noted, we evaluate on the first *1 000* instances from each dataset (after shuffling with a fixed seed of 42). Note that these instances are full sequences containing many tokens; the total number of evaluation tokens is summarized in Table 3. We further verified robustness by repeating selected experiments with substantially larger sample sizes (e.g., 10k instances), and observed identical trends, indicating that our conclusions are insensitive to the particular sample count or ordering.

### A.2.1 MODELS FOR QUALITATIVE SCALING ANALYSIS

**Model.** For the qualitative study of scaling trends, we selected **16 pretrained models** covering a wide range of non-embedding parameter counts, from millions to tens of billions. The models, listed in order of increasing non-embedding parameter counts, are: `pythia-14m`, `distilgpt2`, `gpt2`, `pythia-160m`, `pythia-410m`, `qwen2.5-0.5B`, `opt-1.3b`, `gpt2-xl`, `llama-3.2-1B`, `qwen2.5-3B`, `llama-3.2-3B`, `mistral-7B-v0.1`, `opt-6.7b`, `llama-2-13b-hf`, `qwen2.5-14B`, `llama-2-70b-hf`. This selection spans 14M to 70B models, enabling qualitative analysis of scaling behaviors across different parameter regimes.

### A.2.2 Models for Quantitative Scaling Analysis

**Model.** For the quantitative scaling experiments, we evaluate **over 30 pretrained models**, ranging from **14M to 70B parameters**. All parameter counts are reported as *non-embedding parameters*, i.e., excluding the token embedding layers. These models cover diverse families including GPT-2 (Radford et al., 2019), OPT (Zhang et al., 2022), Pythia (Biderman et al., 2023), LLaMA (Touvron et al., 2023), Mistral (Jiang et al., 2023), and Qwen(Hui et al., 2024). Table 5 summarizes the full set of models and their parameter scales.

| family | model | params | family | model | params |
|---|---|---|---|---|---|
| **gpt-1/2** | openai-gpt | 85,054,464 | **pythia** | pythia-14m | 1,189,888 |
| | gpt2 | 85,056,000 | | pythia-31m | 4,739,072 |
| | gpt2-medium | 302,311,424 | | pythia-70m | 18,915,328 |
| | gpt2-large | 708,390,400 | | pythia-160m | 85,056,000 |
| | gpt2-xl | 1,475,561,600 | | pythia-410m | 302,311,424 |
| **opt** | opt-125m | 85,056,000 | | pythia-1.4b | 1,208,602,624 |
| | opt-350m | 303,357,952 | | pythia-2.8b | 2,517,652,480 |
| | opt-1.3b | 1,208,602,624 | | pythia-6.9b | 6,444,163,072 |
| | opt-2.7b | 2,517,652,480 | **llama** | llama-3.2-1b | 973,146,112 |
| | opt-6.7b | 6,444,163,072 | | llama-3.2-3b | 2,818,747,392 |
| **qwen** | qwen2.5-0.5b | 357,898,112 | | llama-7b | 6,476,271,616 |
| | qwen2.5-1.5b | 1,310,340,608 | | meta-llama-3-8b | 6,979,588,096 |
| | qwen2.5-3b | 2,774,773,760 | | llama-2-13b-hf | 12,688,184,320 |
| | qwen2.5-7b | 6,525,621,760 | | llama-2-70b-hf | 68,452,360,192 |
| | qwen2.5-14b | 13,212,898,304 | **mistral** | mistral-7b-v0.1 | 6,979,588,096 |
| | qwen2.5-72b | 70,214,787,072 | **others** | distilgpt2 | 42,528,768 |

Table 5: Full list of evaluated models and their non-embedding parameter counts.

## B Extensive Training Dynamics Experiments

In Figure 9, we illustrate how cross-entropy and its decomposed components evolve throughout training. The plots cover three datasets and track training progress on Pythia-12B, one of the largest open-source models with available checkpoints. This confirms our main results by showing that all three components are indeed optimized during training.

## C Extensive Qualitative Analysis

In Figure 10, we report extended qualitative results for cross-entropy and its decomposed components across model scales. This figure uses the same set of 32 models as in our main quantitative analysis and further includes Meta-Llama-3.1-405B, one of the largest open-source language models currently available.

The overall trends are fully consistent with our main results. Both cross-entropy and Error-Entropy exhibit clear scaling behavior, with Error-Entropy lying systematically closer to the fitted power-law line. In contrast, the Self-Alignment and Confidence terms do not show a stable decreasing pattern with model size and display much larger variance. Importantly, adding the 405B-parameter model does not change these qualitative conclusions, reinforcing our claim that Error-Entropy is the robust component follows a scaling law in the open-model regime we can study.

## D Derivation of the compensated loss gradient

We collect here the notation and give a detailed derivation of the gradient for the compensated loss

$$\mathcal{L}_\lambda = \mathrm{CE} + \lambda \cdot \mathrm{CONF}, \quad 0 < \lambda < 1. \tag{12}$$

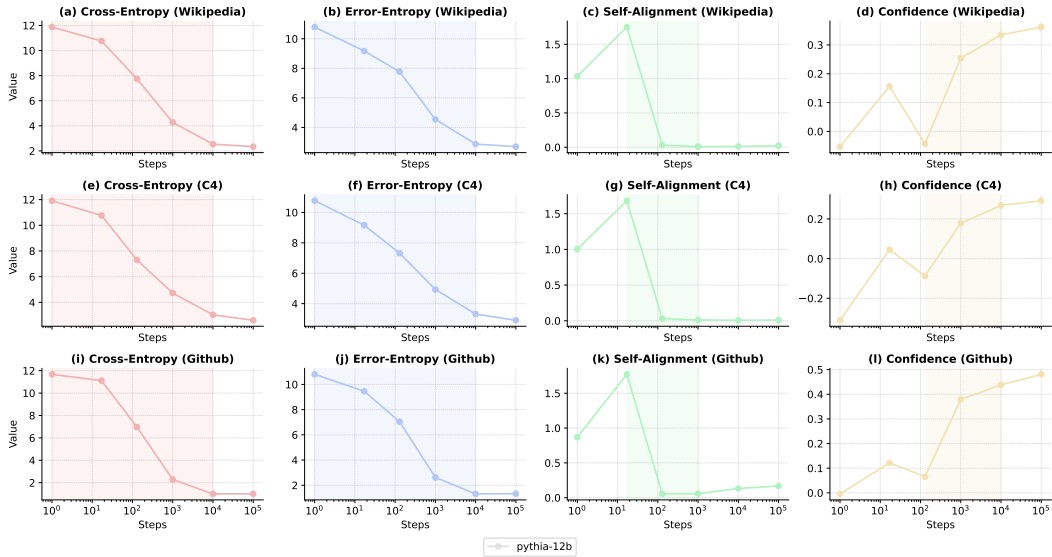

Figure 9: Evolution of cross-entropy and its components during training across three datasets with Pythia-12B.

**Notation.** We consider a corpus with $N$ evaluation tokens indexed by $i = 1, \ldots, N$. For each token $i$: $s_i$ denotes the softmax score of the ground-truth token, $e_i$ denotes its rank-based error level.

For each error level $e$, we group tokens via $I_e = \{i : e_i = e\}, n_e = |I_e|$, and define the error distribution $p_e = n_e/N$ and the *geometric mean* score $Q_e = \left(\prod_{i \in I_e} s_i\right)^{1/n_e}$. We then aggregate over error levels by $C = \sum_e Q_e$ and $q_e = Q_e/C$,

In this notation, the corpus-level cross-entropy and Confidence terms are

$$\mathrm{CE} = -\frac{1}{N} \sum_{i=1}^{N} \log s_i, \tag{13}$$

$$\mathrm{CONF} = \log C = \log \sum_e Q_e. \tag{14}$$

**Derivation.** We treat the scores $\{s_i\}$ as independent variables.

**Cross-entropy term.** The derivative of cross-entropy with respect to $s_i$ is

$$\frac{\partial \mathrm{CE}}{\partial s_i} = -\frac{1}{N} \cdot \frac{1}{s_i} = -\frac{1}{N s_i}. \tag{15}$$

**Confidence term.** For the geometric mean $Q_e$ we have

$$\log Q_e = \frac{1}{n_e} \sum_{j \in I_e} \log s_j. \tag{16}$$

Thus, for a token $i$ with error level $e = e_i$,

$$\frac{\partial \log Q_e}{\partial s_i} = \frac{1}{n_e} \cdot \frac{1}{s_i}, \qquad \Rightarrow \qquad \frac{\partial Q_e}{\partial s_i} = Q_e \cdot \frac{1}{n_e s_i}. \tag{17}$$

Since $C = \sum_e Q_e$, only the term at $e = e_i$ depends on $s_i$, so

$$\frac{\partial C}{\partial s_i} = \frac{\partial Q_{e_i}}{\partial s_i} = \frac{Q_{e_i}}{n_{e_i} s_i}. \tag{18}$$

Using $\mathrm{CONF} = \log C$, we obtain

$$\frac{\partial \mathrm{CONF}}{\partial s_i} = \frac{1}{C} \cdot \frac{\partial C}{\partial s_i} = \frac{Q_{e_i}}{C \, n_{e_i} s_i} = \frac{q_{e_i}}{n_{e_i} s_i}, \tag{19}$$

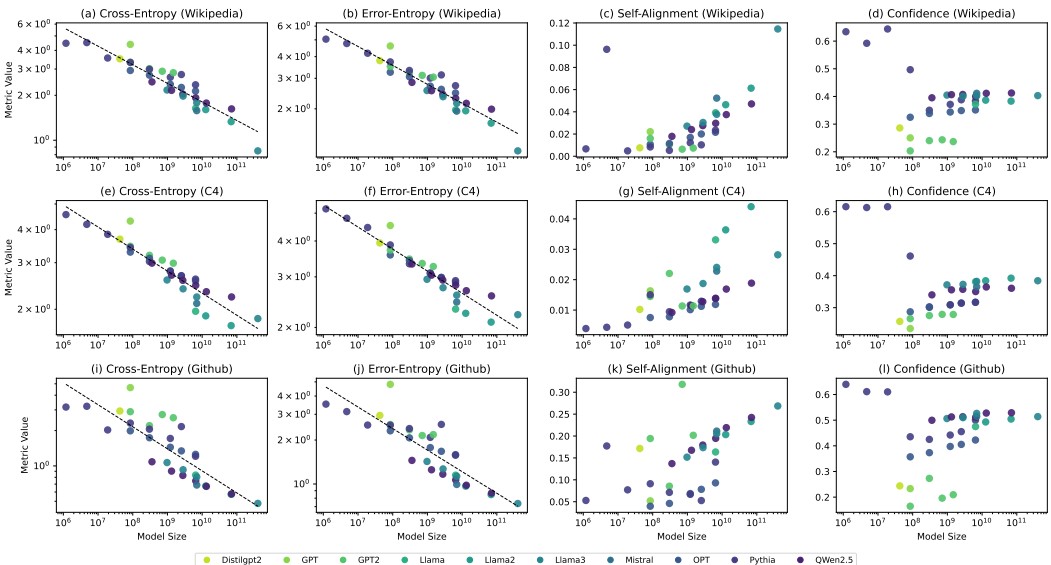

Figure 10: Extended illustration of cross-entropy and its decomposed components over model size.

where we used $q_e = Q_e/C$. Recalling that $p_e = n_e/N$ so that $n_e = Np_e$, we can rewrite this as

$$\frac{\partial \text{CONF}}{\partial s_i} = \frac{q_{e_i}}{Np_{e_i}s_i} = \frac{1}{Ns_i} \cdot \frac{q_{e_i}}{p_{e_i}}. \tag{20}$$

**Compensated loss.** Combining equation 15 and equation 20, the derivative of $\mathcal{L}_\lambda = \text{CE} + \lambda \cdot \text{CONF}$ with respect to $s_i$ is

$$\frac{\partial \mathcal{L}_\lambda}{\partial s_i} = \frac{\partial \text{CE}}{\partial s_i} + \lambda \frac{\partial \text{CONF}}{\partial s_i} \tag{21}$$

$$= -\frac{1}{Ns_i} + \lambda \cdot \frac{1}{Ns_i} \cdot \frac{q_{e_i}}{p_{e_i}} \tag{22}$$

$$= -\frac{1}{Ns_i}\Big(1 - \lambda \frac{q_{e_i}}{p_{e_i}}\Big). \tag{23}$$

Finally, since each score $s_i$ is a smooth function of the logits via the softmax transformation, the compensated loss $\mathcal{L}_\lambda$ is differentiable with respect to the logits, and can be optimized with standard gradient-based training.

# E   STATEMENT ON THE USE OF LARGE LANGUAGE MODELS

We used large language models (LLMs) solely for polishing the writing of this manuscript, including grammar correction, style harmonization, and minor sentence-level rewrites. All technical content—definitions, theorems, methods, experiments, and conclusions—was authored and verified by the authors. Any LLM-suggested phrasing that was adopted was manually reviewed and edited. No unpublished data, proprietary code, or sensitive information was provided to any LLM. This usage does not affect the reproducibility of our methods or results. This disclosure follows the ICLR 2026 policy on LLM usage.

