# OpenReview forum: "What Scales in Cross-Entropy Scaling Law?"
_ICLR.cc/2026/Conference — ICLR 2026 Poster_

### Official Review · Reviewer_cnon · 2025-10-21

**Soundness:** 3
**Presentation:** 3
**Contribution:** 2
**Rating:** 2
**Confidence:** 4

**Summary:**

This paper investigates why the widely used cross-entropy scaling law—which predicts that cross-entropy loss decreases as a power law with model size—begins to fail for very large language models. The authors hypothesize that cross-entropy itself does not truly scale, but rather only a hidden component within it does. To uncover this, they introduce a novel mathematical decomposition of cross-entropy into three distinct parts: Error-Entropy, Self-Alignment, and Confidence. Using a new metric called Rank-Based Error (RBE)—the rank position of the correct token—the paper expresses cross-entropy as the sum of the entropy of the RBE distribution (Error-Entropy), the KL divergence between that distribution and the model’s score distribution (Self-Alignment), and a confidence term related to score normalization. This decomposition offers a more interpretable view of what cross-entropy measures in language model training.

Through extensive experiments across multiple datasets (Wikipedia, C4, and GitHub) and 32 models spanning five orders of magnitude in size, the authors find that only Error-Entropy consistently follows a clean power-law scaling with model size, while the other two components remain mostly invariant. They therefore propose an Error-Entropy Scaling Law as a more fundamental description of model behavior. The paper argues that this new law explains why the classical cross-entropy scaling law appears valid for small models but breaks down for large ones—because Error-Entropy dominates at small scales but shrinks in relative importance as models grow. The authors conclude that Error-Entropy may provide a more accurate and theoretically meaningful measure for understanding and predicting large model performance.

**Strengths:**

1. The paper’s most significant strength is its conceptual novelty.  By
decomposing cross-entropy into Error-Entropy, Self-Alignment, and
Confidence, it provides a new interpretive framework for understanding
what cross-entropy actually measures in language model training. This
move reframes the long-standing “scaling law” discussion away from
raw loss values toward a more structured, information-theoretic
understanding of model behavior.



2. The work offers a coherent explanation for an empirical puzzle that
has troubled practitioners: why the cross-entropy scaling law fits
well for small models but fails at very large scales.  By showing that
only Error-Entropy continues to follow a power law, the paper proposes
a simple and intuitive reason — as models grow, non-scaling
components (Self-Alignment and Confidence) dominate.  This insight
gives practical and theoretical clarity to the field’s observed
irregularities.



3. The decomposition is mathematically clean and transparent.  Starting
from the standard definition of cross-entropy, the authors derive the
decomposition step by step (Eq. 5–8, p.5), connecting it to
measurable quantities like rank distributions and score norms.  This
clear derivation enhances reproducibility and provides a framework
that can be extended or critiqued rigorously by others.



4. The paper demonstrates solid empirical design: it tests the hypothesis
across 32 language models (Pythia, GPT-2, Qwen, LLaMA, OPT, Mistral,
etc.) and three distinct corpora (Wikipedia, C4, GitHub) over five
orders of magnitude in scale.  Although limited in token sampling,
this breadth of architectures strengthens the empirical robustness and
helps generalize the scaling claim within the text domain.


5. Finally, the findings have broad implications.  Practically,
recognizing that only Error-Entropy scales could inform training
objectives and compute allocation for large models.  Theoretically,
the decomposition opens a potential bridge between rank-based
evaluation and information theory, offering a new direction for
research on learning dynamics and interpretability in large-scale
neural systems.

**Weaknesses:**

1. Although the decomposition is mathematically valid (Eq. 8, p. 5), it
is fundamentally algebraic rather than theoretical.  The decomposition
introduces new quantities (RBE), but this does not derive them from a
principled model of language or probability theory. In other words, “Error-Entropy” is justified empirically, not from first principles
— there is no information-theoretic or Bayesian derivation explaining
why the entropy of rank distribution should obey a power law. Consequently, the argument is more descriptive than explanatory: it simply
re-expresses cross-entropy in new coordinates rather than uncovering
causal structure. In short: the authors show that “error-entropy scales,” but not why
it should do so.


2. The central variable, Rank-Based Error (RBE), depends on sorting model
outputs by rank (p. 4 – 5).  While this makes intuitive sense (rank as semantic accuracy), it has drawbacks.  Primarily,  ranking operations are non-differentiable. Hence, RBE and its entropy cannot directly serve as a training objective; the authors
mention this only implicitly. Rank distributions depend on vocabulary size.  This can distort entropy values when models differ in
tokenization or vocabulary. Thus, “error-entropy” may not generalize well across architectures
or token sets.



3. The paper shows strong empirical correlations (R^2 being 0.97 ),
but: it does not test alternative decompositions — e.g., whether
other monotonic functions of cross-entropy could produce similar
scaling. It provides no ablation or causal test: does directly minimizing
error-entropy improve model scaling?  (They hint at this as “future
work.”)


4. Despite spanning “five orders of magnitude,” the experiments rely
mostly on open-weight transformer families (Pythia, Qwen, GPT-2,
LLaMA, etc.) trained on text-only datasets (Wikipedia, C4, GitHub;
p. 7 – 8). All datasets share similar linguistic distributions.  The sample size
per dataset is small (≈ 1 000 tokens per model; App. A.2 p. 12),
meaning statistical noise could bias scaling exponents. Hence, the “Error-Entropy Scaling Law” may not generalize beyond
their curated setup.



5. In Table 1 and 2 (p. 7 – 8), the differences in R^2 and |\delta| between
cross-entropy and error-entropy are numerically modest (e.g., 0.973 →
0.975).  Yet the paper interprets these small gains as evidence of a
fundamentally new law.  Given log–log regression’s sensitivity to
outliers and limited sample size, the improvement could easily arise
from noise or fitting artefacts.

**Questions:**

1. Error-Entropy is defined using rank statistics, which are inherently
non-differentiable. This raises an essential question:
Can Error-Entropy (or a differentiable surrogate) be used as a
practical training objective or evaluation metric?  If not, what is
its functional utility beyond post-hoc analysis? The paper hints (p.9) that Error-Entropy could inspire new loss
functions, but provides no method for computing gradients or
approximations.  Bridging this gap between theoretical definition and
practical optimization would determine whether this concept can
actually influence model development.



2. The paper’s key claim is that Error-Entropy—the Shannon entropy of
the rank-based error (RBE) distribution—is the true quantity that
scales.  However, is Error-Entropy revealing a genuinely new
theoretical property of learning dynamics, or is it simply a
re-expression of cross-entropy under a rank-based transformation? If Error-Entropy’s scaling behavior arises purely from the algebraic
decomposition of cross-entropy, it may not have independent
explanatory power.  Clarifying whether this metric corresponds to a
deeper principle (e.g., a conserved information quantity or
energy-like invariant) is crucial for establishing its theoretical
legitimacy.

---

> ### Author Response · Authors · 2025-11-28
>
> Dear Reviewer cnon,
>
> Thank you for your careful reading and extensive comments on our work. We appreciate your recognition of the conceptual novelty of our proposed  cross-entropy decomposition, as well as your positive remarks on our mathematical clarity and experimental breadth.
>
> Regarding your concerns:
>
> ### Q1: Can Error-Entropy (or a differentiable surrogate) be used as a practical training objective or evaluation metric? If not, what is its functional utility beyond post-hoc analysis? The paper hints (p.9) that Error-Entropy could inspire new loss functions, but provides no method for computing gradients or approximations. Bridging this gap between theoretical definition and practical optimization would determine whether this concept can actually influence model development.
>
> Thank you for this important question. We agree that connecting Error-Entropy (EE) to practical optimization is crucial.
>
> EE is rank-based and therefore not differentiable with respect to the logits, which makes it difficult to optimize directly. However, our decomposition
> $$
> \text{CE} = \text{EE} + \text{SA} - \text{CONF}
> $$
> suggests a practical workaround.
> Empirically, we find that EE is the dominant term in CE across model scales.
> This suggests that SA and CONF can be treated as residual components relative to EE. Among them, SA has consistently low magnitude, while CONF remains a substantial part of CE, as shown in Figure 8. Importantly, CONF is fully differentiable.This allows us to construct a surrogate loss by removing or penalizing CONF, which implicitly shifts optimization toward EE:
> $$
> \mathcal{L}_\lambda = \text{CE} + \lambda \cdot \text{CONF}, 0<\lambda<1.
> $$
>
> In our notation, CE can be written as $-\frac{1}{N}\sum_i \log s_i$, where $s_i$ is the $i$-th example's score, and CONF is $\log \sum_e Q_e$, where $Q_e$ represents the mean scores over errors. Differentiating $\mathcal{L}_\lambda$ with respect to $s_i$ yields
>
> $$
> \frac{\partial \mathcal{L}_\lambda}{\partial s_i}
> = -\frac{1}{s_i}\Big(\frac{1}{N} - \lambda \frac{Q_e}{Cn_e}\Big)
> = -\frac{1}{N s_i}\Big(1 - \lambda \frac{q_e}{p_e}\Big),
> $$
>
> where $p_e$ denotes the error distribution and $C$ is the normalizing constant.
> Intuitively, the factor $\big(1 - \lambda \frac{q_e}{p_e}\big)$ reduces the gradient pressure that comes from overly improving $q_e$, and instead puts relatively more emphasis on adjusting the model so that $p_e$ itself improves. In other words, we can bias learning away from over-optimizing Confidence and more toward the part of CE that actually scales (EE), while staying fully within standard backpropagation.
>
> Beyond such surrogates, EE can also be used as a non-differentiable objective in RL-style fine-tuning, where it appears effectively as a reward signal, without requiring gradients through ranks.
>
> We thank the reviewer for highlighting this important point. To address it, we have added a new Section 6.2 on our revised paper discussing EE-inspired training objectives.

---

> ### Author Response · Authors · 2025-11-28
>
> ### Q2: Is Error-Entropy revealing a genuinely new theoretical property of learning dynamics, or is it simply a re-expression of cross-entropy under a rank-based transformation? If Error-Entropy’s scaling behavior arises purely from the algebraic decomposition of cross-entropy, it may not have independent explanatory power. Clarifying whether this metric corresponds to a deeper principle (e.g., a conserved information quantity or energy-like invariant) is crucial for establishing its theoretical legitimacy.
>
> Thank you for this thoughtful question. We believe that Error-Entropy is not just a re-expression of cross-entropy, but corresponds to a deeper principle: it captures the information of the model’s **error distribution**.
>
> In fact, the notion of error-entropy is not new. In the Information Theoretic Learning (ITL) field, it has been extensively studied and is known to enjoy many desirable properties (e.g., robustness under heavy-tailed, non-Gaussian noise). However, this line of work is entirely restricted to **regression** and small-scale non-parametric settings: the model outputs a real value, there is a natural error $prediction − target$, and learning is formulated on error distribution, not probabilities. These assumptions do not match the **classification** setup of LLMs, where the model outputs a probability distribution over tokens and the main question is “where does the correct token rank among all candidates?”. As a result, existing EE results cannot be directly applied to language models.
>
> Our algebraic decomposition is precisely what bridges this gap. Starting from standard cross-entropy in language modeling, we (i) define an **rank-based error** for LLMs, which restores an “error-distribution” object in the classification setting, and (ii) separate this from how the model encode that difficulty in probabilities. Error-Entropy is then the entropy of the RBE distribution, directly aligned with the ITL notion, while other components describe how the probability scores are organized on error. In this way, the decomposition is not a simple change on CE. It provides a important connection from CE-based LLM to the ITL perspective on error distributions, and empirically shows that this EE is the one that actually scales.
>
> On top of this foundation, future work can both (a) import ITL optimization techniques to design EE-guided training objectives for LLMs, and (b) draw on ITL theory to further explain why the error-entropy exhibits the observed scaling behavior. We thank the reviewer for highlighting this important point and have added a new Section 6.1 on our revised paper about this.
>
> ### W1：The reviewers think we only show that "error-entropy scales," but not why it should do so.
>
> This concern is closely related to Q2, and we address it in more detail there. Briefly, while our decomposition is indeed an exact algebraic identity, its role is not merely rewriting CE in another form. It isolates a rank-based error-entropy term (EE) that (i) is closely connected to the Information-Theoretic Learning literature on minimum error entropy, and (ii) empirically shows robust power-law scaling, in contrast to the other two components.
>
> We believe our work takes a first step in this direction: instead of treating CE as a black box, we show that its apparent scaling can be understood as the scaling of Error-Entropy. Building on its connection to ITL-style error entropy, we believe future work will be better positioned to develop a full first-principles theory of why CE (and EE) scale, ultimately providing a more complete answer to the scaling-law question.
>
> ### W2: RBE and its entropy cannot directly serve as a training objective. "Error-entropy" may not generalize well across architectures or token sets.
>
> Thank you for your concern about the practical use of EE as a training objective. This overlaps with Q1, where we discuss in detail how the decomposition suggests practical, differentiable surrogates (via the Confidence term).
>
> Regarding the dependence on vocabulary size and tokenization, we would like to emphasize that this issue is not specific to EE. Standard cross-entropy itself depends on the tokenizer and vocabulary. In our experiments, EE is always evaluated under exactly the same setup as CE: on the same corpora, with the same tokenization, and using the same model logits. Scaling laws for EE and CE are fitted following the usual practice in CE scaling-law studies. Thus, our study is not subject to additional assumptions or limitations beyond those already in CE. We therefore do not view vocabulary dependence as undermining our main conclusions; rather, EE demonstrates more robust scaling within the same constraints under which CE scaling laws are typically considered meaningful.

---

> ### Author Response · Authors · 2025-11-28
>
> ### W3: The reviewer think the study does not test alternative decompositions. It provides no ablation or causal test: does directly minimizing error-entropy improve model scaling?
>
> We appreciate the reviewer’s suggestion to explore alternative decompositions. First, to the best of our knowledge, we are the first to provide a concrete decomposition of cross-entropy for LLMs into components with clear semantics. Previous LLM work has mostly treated cross-entropy as a black-box, while our decomposition separates the part that measures how well the model ranks the correct token (Error-Entropy) from the parts that describe how this difficulty is reflected in probabilities (Self-Alignment and Confidence). This makes the behavior of CE much more interpretable. Second, Error-Entropy in our formulation aligns closely with the notion of error entropy studied in Information-Theoretic Learning, which gives a direct and principled connection between CE-based language modeling and an established theoretical framework. Finally, other decompositions may be explored in future work, but the specific interpretability and the deeper theory connection we emphasize here are distinctive contributions of our decomposition.
>
> Regarding "does directly minimizing Error-Entropy improve model scaling?", we agree this is an important research goal and label it as future work. As discussed in Q1, our decomposition already suggests resonable EE-inspired objectives, indicating that such studies are technically attainable. However, we believe a full optimization study is beyond the scope of this paper and does not affect the validity of our current conclusions.
>
> ### W4: All datasets share similar linguistic distributions. The sample size per dataset is small (≈ 1 000 tokens per model; App. A.2 p. 12), meaning statistical noise could bias scaling exponents.
>
> We apologize for the confusion caused by our description of the experimental setup. First, our datasets are not "all similar linguistic distributions": besides Wikipedia and C4, which are standard but distinct natural-language corpora, Pile-GitHub is dominated by source code and code-related text, which has very different statistical structure. Our goal is mainly to study text-domain LMs, and within this domain we already span different distributions.
>
> Second, we do not use "1,000 tokens per model". We use about 1,000 **instances** per test, and each instance is actually a full document with many tokens. Therefore, our evaluation sample size is actually large. The actual token counts used are:
>
> |        | Wikipedia | C4      | Pile-GitHub |
> | ------ | --------- | ------- | ----------- |
> | Tokens | 704,185   | 417,140 | 1,091,481   |
>
> Third, we mentioned robustness to sample size (in App. A.1: Sampling protocol.): we repeated experiments with larger samples (10k instances) and observed essentially identical trends. For example, on Wikipedia with 10k instances, the $R^2$ values are:
>
> |        | CE     | EE         | SA     | CONF   |
> | ------ | ------ | ---------- | ------ | ------ |
> | Qwen   | 0.9871 | **0.9879** | 0.9850 | 0.5855 |
> | Pythia | 0.9588 | **0.9796** | 0.0012 | 0.7868 |
> | GPT2   | 0.9625 | **0.9744** | 0.8579 | 0.6491 |
> | All    | 0.8509 | **0.8744** | 0.2734 | 0.0239 |
>
> These numbers differ only slightly from those in the main paper and lead to the same qualitative conclusion: Error-Entropy is the only component that exhibits stable scaling, while Self-Alignment and Confidence do not. This shows that our findings are not an result of small sample size and in fact robust.
>
> ### W5: In Table 1 and 2 (p. 7 – 8), the differences in R^2 and |\delta| between cross-entropy and error-entropy are numerically modest (e.g., 0.973 → 0.975).
>
> We appreciate the concern and would like to clarify two key points that may not have been fully addressed in the review.
>
> First, our conclusion is based on a process of elimination. We decompose cross-entropy into three components: Error-Entropy (EE), Self-Alignment (SA), and Confidence. Both theory and empirical results show that **SA and Confidence do not exhibit stable scaling behavior**. Given that CE is exactly the sum of these three terms, EE must be the component that carries the robust scaling.
>
> Second, we note that EE currently dominates the total cross-entropy, which makes the scaling behavior of CE and EE appear similar at present scales. However, as models grow larger and the relative share of EE shrinks (as shown in Figure 8), we expect the power-law fit of cross-entropy to become less accurate, while EE continues to follow a stable trend. This suggests that EE captures the part of CE that genuinely scales with model size, and may become an even more reliable predictor in future regimes.
>
> *Thank you again for your time and constructive feedback. We hope our responses have addressed your concerns, and we kindly request you to consider raising the rating of our paper if you find our responses satisfactory.*

---

### Official Review · Reviewer_Vend · 2025-10-30

**Soundness:** 3
**Presentation:** 3
**Contribution:** 4
**Rating:** 8
**Confidence:** 4

**Summary:**

This paper investigates the widely observed phenomenon that the cross-entropy scaling law, which predicts that model loss decreases as a power-law of model size, begins to break down at larger scales. The authors hypothesize that the root cause is that cross-entropy loss itself is not the quantity that truly scales. To test this, they introduce a novel and exact mathematical decomposition of the cross-entropy loss into three interpretable components: Error-Entropy, Self-Alignment, and Confidence. Through extensive experiments on 32 models spanning five orders of magnitude (up to 70B parameters), they find that only Error-Entropy, which measures the entropy of the rank of the correct token, exhibits a robust and predictable power-law scaling. The other two components remain largely invariant or noisy. The paper argues that the apparent scaling of cross-entropy in smaller models is an illusion caused by Error-Entropy being the dominant component. As models grow larger, its relative contribution diminishes, allowing the non-scaling components to disrupt the overall power-law trend. The authors propose the "Error-Entropy Scaling Law" as a more fundamental principle for understanding and predicting model performance.

**Strengths:**

1. The core idea of decomposing the cross-entropy loss is highly original. The three proposed components are not only mathematically sound but also intuitively interpretable. They map to distinct aspects of a model's learning process: getting the answer right (Error-Entropy), calibrating its output probabilities to its own error patterns (Self-Alignment), and expressing certainty by assigning low scores to incorrect tokens (Confidence). This provides a powerful new analytical tool.

2. The paper offers a simple and elegant explanation for a complex and important problem—the breakdown of cross-entropy scaling laws. The finding that only one component truly scales, and that its proportion changes with model size (Figure 8), compellingly explains why the law holds for smaller models but falters for larger ones.

3. The discovery of the Error-Entropy Scaling Law could have broad implications. It suggests a more reliable metric for extrapolating model performance and could inspire new training objectives focused directly on minimizing Error-Entropy, potentially leading to more efficient training or better-calibrated models.

**Weaknesses:**

1. The paper is motivated by the failure of scaling laws in the very large model regime, yet the analysis stops at 70B parameters. The most dramatic and puzzling scaling behaviors are often observed in models exceeding 100B parameters. Without data from these frontier models, the claim that the Error-Entropy law holds true at the absolute largest scales remains an extrapolation.

2. The analysis of how the components evolve during training (Figures 3-6) is based on the Pythia model family, with the largest model being 1B parameters. While insightful, it's unclear if these same dynamics (e.g., Error-Entropy being optimized first) hold true when training much larger models (e.g., 70B), where training dynamics can be qualitatively different.

3. The study exclusively analyzes existing pre-trained checkpoints. It does not explore how different training configurations (e.g., optimizers, data quality, learning rate schedules) might influence the behavior and proportions of the three components. It's possible that certain training strategies could alter the dynamics of Self-Alignment or Confidence, affecting the overall cross-entropy behavior.

**Questions:**

How many tokens are used to train the models in appendix A

---

> ### Author Response · Authors · 2025-11-28
>
> Dear Reviewer Vend,
>
> Thank you for your constructive feedback and positive assessment of our work. We appreciate your recognition of our new analytical framework for understanding cross-entropy scaling laws and its broad potential implications in the field of LLMs.
>
> Regarding your concerns:
>
> ### W1: Lack of data from frontier models exceeding 100B parameters
>
> Thank you for this important observation. To address this concern, we extend our study to Meta-Llama-3.1-405B in the revised version, one of the largest open-source models to date, and report the results in the appendix. The results remain consistent with our expectations: SA and Confidence continue to show no clear scaling behavior, while EE exhibits more stable scaling than CE, with more stable exponents and higher $R^2$.
>
> We fully agree that confirming the Error-Entropy Scaling Law at frontier scales beyond current open models is an open and exciting problem. Existing evidence from GPT-4–scale models [1] suggests that noticeable offsets of cross-entropy appear only in the trillion-parameter setting. Testing whether EE continues to scale cleanly there is a important direction for future work.
>
> ### W2: Training Dynamics might be different for much larger models (e.g. 70B).
>
> Thank you for raising this point. To address your concern, we additionally analyzed the full training dynamics of Pythia-12B, which is the largest open-source model family for which intermediate checkpoints are publicly available. We include the result in the revised paper, and it shows that, across three orders of magnitude (160M-12B), the evolution of CE, EE, SA, and ConF over training steps follows the similar pattern. These suggest that the training dynamics described in our paper are not an artifact of very small models, but remain stable over a wide range of sizes. We then reasonabley expect the same pattern to extend to larger models such as 70B.
>
> ### W3: Different training configurations (e.g., optimizers, data quality, learning rate schedules) might influence the behavior and proportions of the three components.
>
> Thank you for pointing out the potential impact of different training configurations. Our focus in this work is aligned with the classic scaling law: we study the behavior of well-trained models, as a function of model size, rather than the details of the optimization path.
>
> From this perspective, our decomposition and conclusions rely only on the final checkpoints. Moreover, the model families we analyze (GPT-2, Pythia, Qwen, LLaMA, OPT, Mistral, etc.) already span a wide range of training recipes in practice. Despite this implicit diversity, we observe that (i) EE follows a clean scaling law, and (ii) Self-Alignment and Confidence do not exhibit systematic scaling, across datasets and model families.
>
> We agree that controlled studies under different training settings would be a valuable extension.
> Such work could offer complementary insights into how the Error-Entropy scaling behavior emerges.
> We view this as a parallel direction that builds on our findings and can be explored in future work.
>
> ### Q1: How many tokens are used to train the models in appendix A?
>
> Thank you for asking for this clarification. In Appendix A.1, we state that we evaluate each model on 1,000 instances per dataset. Here, "instance" refers to a full document. Since we compute next-token predictions for every position in each instance, each evaluation instance corresponds to hundreds of token predictions. For the datasets we evaluate in Appendix A, the total number of tokens is as follows and we will clarify these numbers in the revised appendix:
>
> |           | Wikipedia | C4      | Pile-GitHub |
> |-----------|-----------|---------|-------------|
> | Tokens    | 704,185   | 417,140 | 1,091,481   |
>
> In addition, we also repeated experiments with 10k instances; For example, on Wikipedia with 10k instances, the $R^2$ values are very close to those in the main paper and do not deviate from our primary results:
>
> |        | CE     | EE         | SA     | CONF   |
> | ------ | ------ | ---------- | ------ | ------ |
> | Qwen   | 0.9871 | **0.9879** | 0.9850 | 0.5855 |
> | Pythia | 0.9588 | **0.9796** | 0.0012 | 0.7868 |
> | GPT2   | 0.9625 | **0.9744** | 0.8579 | 0.6491 |
> | All    | 0.8509 | **0.8744** | 0.2734 | 0.0239 |
>
> [1] Achiam, Josh, et al. "GPT-4 Technical Report." arXiv:2303.08774 (2023).
>
> *Thank you for your suggestions. We hope our responses clarify the issues raised, and we kindly request you to consider raising the rating of our paper if you find our responses satisfactory.*

---

### Official Review · Reviewer_VUMH · 2025-11-01

**Soundness:** 2
**Presentation:** 3
**Contribution:** 3
**Rating:** 6
**Confidence:** 3

**Summary:**

The paper decomposes cross-entropy into three components—Error Entropy (EE), Self-Alignment, and Confidence. It experimentally shows that the part that actually scales with model size is Error Entropy. Based on this, it proposes an explanation for why the cross-entropy scaling law fails at very large model sizes: as the model gets larger, the proportion of cross-entropy contributed by Error Entropy becomes smaller.

**Strengths:**

1. The overall structure is clear, the figures are appropriate, and readers can easily grasp the core ideas.
2. The mathematical derivations are clear and rigorous.
3. It offers a new perspective on the cross-entropy scaling law, which is thought-provoking.

**Weaknesses:**

1. Experimental results are not strong enough.
   The authors aim to use Error Entropy to explain the failure of the cross-entropy scaling law at large parameter sizes. However, possibly due to the experimental setup or model size, the experiments in the paper do not clearly demonstrate the failure of the scaling law. In this situation, relying only on EE having a slightly higher \(R^2\) than CE (and not even for all model x dataset combinations) to argue that EE is the part that actually scales is not very convincing.
   If the authors could change the experimental setting (e.g. try different datasets, context lengths, or model sizes) to actually reproduce the failure of the CE scaling law, and at the same time show that EE still fits well, then the results would be much more persuasive.

2. Confusing phrasing.
  While the overall argument is clear and the main findings are well laid out, a few parts are awkwardly expressed. For instance, in Section 3.3 on Self-Alignment, lines 296–303 already state clearly that Self-Alignment is about aligning \(p_e\) and \(q_e\). However, the two sentences that follow (lines 304–308) are hard to follow. - “Since different models have different error distributions, they also produce different probability scores.” — this seems to claim a causal link between error distributions and probability scores, but the paper does not substantiate that claim (correlation does not imply causation). - “If, by contrast, probabilities truly represented the single real language distribution as suggested by cross-entropy, we would not expect variation in probability scores across models, which deviates from empirical observations.” — this is also unclear, because a model’s probability distribution can vary for many ordinary reasons (initialization, training dynamics, local optima, etc.), so variation across models is not in itself evidence against cross-entropy. If I am misunderstanding the authors’ intent here, clarification would help — the current wording leaves too much room for interpretation.

**Questions:**

1. In Table 1, GPT-2’s CE \(R^2\) on the GitHub dataset is only 0.0717, which is far from the other results. Is this a typo?
2. Can you provide a possible explanation for why the share of EE in CE decreases as model size increases?
3. Why are the models used in the qualitative analysis different from those in the quantitative analysis? In other words, since the quantitative analysis already covers 30 models, why does the qualitative analysis only show 16?

---

> ### Author Response · Authors · 2025-11-28
>
> Dear Reviewer VUMH,
>
> Thank you for your detailed comments and positive feedback on our work. We are pleased that you found our paper well-organized and easy to follow, and that you consider our perspective on the Error-Entropy Scaling Law thought-provoking.
>
> Regarding your concerns:
>
> ### W1: Experimental results are not strong enough. If the authors could change the experimental setting (e.g. try different datasets, context lengths, or model sizes) to actually reproduce the failure of the CE scaling law, and at the same time show that EE still fits well, then the results would be much more persuasive.
>
> Thank you for raising this point. We agree that, an ideal case would be a setting where CE scaling law clearly collapses while EE continues to follow a clean scaling curve.
>
> To better address this concern, we have extended our analysis in the revised version to include Meta-Llama-3.1-405B, one of the largest open-source models currently available. The results remain consistent with our expectations: Self-Alignment and Confidence still do not exhibit clear scaling behavior, while Error-Entropy continues to follow a more stable power-law compared with CE. This supports our central claim that EE is the component within CE that most robustly reflects model improvements.
>
> In our decomposition, only EE directly reflects the model’s ability to rank the correct token, while SA and CONF describe how this difficulty is encoded into probabilities. There is no reason for these two terms to shrink as the model improves, and empirically we observe that they do not.
> As a result, the scaling behavior of CE becomes less stable as the influence of these non-scaling terms increases with scale. This provides an explanation for the additive offset observed in the GPT-4 report [1], where CE begins to deviate from a clean power law in extremely large models (beyond 1T parameters). Our decomposition suggests that this deviation can be viewed as the growing relative weight of the non-scaling SA and CONF parts.
>
> As a result, the scaling behavior of cross-entropy becomes less stable at large scales, as the relative weight of these non-scaling terms increases. This explains the additive offset observed in the GPT-4 report [1], where CE deviates from a pure power law:
>
> $$
> L(N) = A \cdot \text{Size}^{-\alpha} + B.
> $$
>
> Under our decomposition, the offset \( B \) can be interpreted as the residual effect of SA and CONF that do not scale cleanly with model size.
>
> We agree that constructing more extreme scenarios where CE visibly fails while EE remains clean (e.g., with specially designed data or frontier-scale models) would further strengthen this theory, and we see this as a interesting direction for future work, rather than a limitation of our current claims.
>
> ### W2: About the confusing phrasing.
>
> Thank you for highlighting this paragraph. Our goal in Section 3.3 is to clarify the function and performance of Self-Alignment, not to claim a strong causal link or to argue that cross-entropy is "invalid."
>
> First, the sentence *"Since different models have different error distributions, they also produce different probability scores."* was intended as a summary of an empirical regularity, not as a causal statement. In Figure 6 we show that, for well-trained models, the empirical error distribution $p_e$ and the score distribution $q_e$ are very close. Across models, differences in $p_e$ are systematically accompanied by differences in $q_e$. Self-Alignment is precisely the term that measures how tightly the model’s probability scores track its own error pattern, and our wording was meant to describe this association.
>
> Second, the latter sentence is intended to further motivate the specific role of Self-Alignment. In the standard CE formulation, minimizing CE corresponds to minimizing $\mathrm{KL}(p_{\text{data}} \Vert p_\theta) + H(p_{\text{data}})$. Under this, if multiple differnent models are trained on the same data and fully optimized, they are all trying to approximate the same underlying language distribution $p_{\text{data}}$. Differences in training primarily affect how fast they approach this target, not what the target is. In practice, however, even among well-trained models we observe substantial, structured variation in the output probabilities. Our point is not that this variation contradicts cross-entropy, but that our decomposition makes this variation interpretable: the Self-Alignment term shows that model probabilities encode not only the natural data distribution, but also specific alignment between error patterns and probability scores.
>
> We hope this explanation addresses your concern, and we apologize for the confusion caused by our phrases.
>
> [1] Achiam, Josh, et al. "Gpt-4 technical report." arXiv preprint arXiv:2303.08774 (2023).

---

> ### Author Response · Authors · 2025-11-28
>
> ### Q1: In Table 1, GPT-2’s CE $R^2$ on the GitHub dataset is only 0.0717, which is far from the other results. Is this a typo?
>
> We are glad you noticed this special value; it is not a typo. For clarity, the table below reports the CE and EE values of the GPT-2 family on the Pile-GitHub subset, together with the $R^2$ of the log–log scaling fit:
>
> | Metric | gpt2    | gpt2-medium | gpt2-large | gpt2-xl | $R^2$ (log-param fit) |
> | ------ | ------- | ----------- | ---------- | ------- | --------------------- |
> | CE     | 2.84790 | 2.15778     | 2.66675    | 2.51462 | 0.0717                |
> | EE     | 2.48698 | 2.27442     | 2.10629    | 2.11663 | 0.9262                |
>
> For the older GPT-2 models, the Pile-GitHub subset behaves more like an out-of-distribution (OOD) dataset relative to their original training data. In this situation, CE does not vary monotonically with model size (e.g., GPT-2-medium has much lower CE than other GPT2 model), so a simple power-law fit yields a very low $R^2$. In contrast, EE decreases much more regularly across the same models, which results in a clean scaling fit with $R^2 \approx 0.93$.
>
> This pattern is consistent with our theory: on such OOD data, the model can still retain great ranking ability (captured by EE), but may fail to translate this into a reliable probability distribution over outputs, leading to unstable CE scaling behavior. We appreciate the reviewer drawing attention to this special case; in future work, we plan to train models and perform systematic OOD evaluations to further test and refine this insight.
>
> ### Q2: Can you provide a possible explanation for why the share of EE in CE decreases as model size increases?
>
> Thank you for this question. As we briefly note in lines 345–349, Error-Entropy is the part  that directly reflects whether the model can put the correct token near the top rankings, while Self-Alignment and Confidence are tightly tied to the probability scores themselves, rather than to correctness.
>
> As model size increases, the model’s ability improves substantially, so EE is strongly optimized and decreases clearly with scale. By contrast, the other components do not shrink in the same way: training does not explicitly push |SA|+|CONF| toward zero, and empirically we observe that their sum remains roughly constant or even grows slightly larger. As a result, EE shrinks faster than the other two terms, so the ratio EE/CE naturally decreases as models become larger.
>
> ### Q3: Why are the models used in the qualitative analysis different from those in the quantitative analysis? In other words, since the quantitative analysis already covers 30 models, why does the qualitative analysis only show 16?
>
> Thank you for your question regarding Figure 7. Our quantitative analysis indeed uses all 32 models when fitting scaling laws and reporting statistics. In the qualitative plots, however, we want to give readers a clear visual sense of how CE and its components evolve with model size. So we simply chose a representative subset to show the trend. For completeness, we have included the full results for all 32 models in the revised paper version.
>
> *We appreciate your insightful questions and hope our responses have addressed your concerns. We will incorporate your valuable suggestions into our revised manuscript. If you find our responses satisfactory, we kindly request you to consider raising the rating of our paper.*

---

### Official Review · Reviewer_6E3g · 2025-11-04

**Soundness:** 2
**Presentation:** 3
**Contribution:** 2
**Rating:** 4
**Confidence:** 3

**Summary:**

This paper proposes an alternative rank-based metric, error-entropy, and argues that it has a more robust scaling law than CE. It shows that CE can be decomposed into three components: error-entropy, self-alignment, and confidence, and through experiments, authors demonstrate that only error-entropy scales consistently with model size, while the other components remain random. Since error-entropy becomes a smaller part of CE as model size increases, this can potentially explain why scaling laws fail for larger models.

**Strengths:**

1. Novelty: the rank-based metric, error-entropy, as well as the CE decomposition are novel as far as I know. The error entropy has nice properties, in particular robustness to post-processing such as temperature scaling.

2. Extensive experiments: this paper includes many experiments, covering the training dynamics of smaller models and the scaling of the three components in larger models. There is a variety of models and datasets, including text and coding tasks.

3. Clarity: the paper is well-written and the exposition is clear.

**Weaknesses:**

My main concern is the significance of the findings:
- Is there evidence that using EE, one can predict the EE of a larger model better than using CE to predict CE?
- Is EE as indicative of downstream performance as CE?
- Regarding self-alignment and confidence: from section 3, self-alignment decreases during training and confidence increases, so I would expect self-alignment to be smaller and confidence to be larger as model size increases. However, self-alignment has a clear upward trend as models get larger, and confidence doesn't have a clear trend at all. Are these metrics meaningful in practice? Why would self-alignment increase for larger models that have more capacity for learning?
- In section 5, the paper argues that self-alignment and confidence might be the reason for the CE scaling law to break down at larger model sizes. However, in the scaling plot, it's not clear that there's a CE scaling breakdown and EE has more robust scaling. Can the authors provide evidence that EE doesn't suffer from the same issue?

**Questions:**

See weaknesses

---

> ### Author Response · Authors · 2025-11-28
>
> Dear Reviewer 6E3g,
>
> Thank you for your careful reading and thoughtful reviews. We appreciate your recognition of the novelty of our proposed error-entropy metric and the cross-entropy decomposition, as well as your positive comments on the clarity of our exposition and experimental coverage.
>
> Regarding your concerns:
>
> ### W1: Is there evidence that using EE, one can predict the EE of a larger model better than using CE to predict CE?
>
> Thank you for this interesting question. In fact, this is exactly what our paper is concerned with: whether EE exhibits a more stable scaling behavior, so that it can be used to analyze model performance and to extrapolate to future, larger models.
>
> (i) In standard in-domain settings such as Qwen2.5 on Wikipedia/C4, the CE scaling law is already extremely clean. In this case, both CE and EE can be used to predict larger models with very small error. Our preliminary analysis (Section 4.2) shows that both CE and EE achieve very high and similar $R^2$ when fitting log-linear scaling laws. This indicates that, when CE scaling is well-behaved, EE provides essentially the same predictive power as CE for larger models.
>
> To make this more concrete, we also conducted a simple predict experiment on the Pythia family. We fit a log-linear scaling law on the four smallest models (14M–160M) and then use it to predict the losses of larger models (1.4B–6.9B). For each corpus and metric, we report the relative prediction error $|L_{\text{pred}} - L_{\text{real}}| / L_{\text{real}}$:
>
> | Target model  | CE (Wiki) | EE (Wiki) | CE (C4) | EE (C4) | CE (GitHub) | EE (GitHub) |
> |--------------|-----------|-----------|---------|-----------|-------------|-------------|
> | pythia-1.4b  | 0.0634    | **0.0372**| 0.0230  | **0.0119**| **0.1724**  | 0.1755      |
> | pythia-2.8b  | 0.1843    | **0.1520**| 0.0538  | **0.0244**| 0.4331      | **0.3998**  |
> | pythia-6.9b  | 0.1677    | **0.1317**| 0.1206  | **0.0951**| 0.1890      | **0.1745**  |
>
> The results show that EE almost always predicts more accurate than CE. For both Wikipedia and C4, EE achieves smaller relative prediction errors than CE for all models. On the GitHub corpus, CE and EE are very close at 1.4B, but EE clearly improves the prediction accuracy for the larger models. Taken together, these results support our view that, in well-behaved scaling scenes, EE is at least as good as CE for predicting the behavior of larger models, and often slightly better.
>
> (ii) More interestingly, in some biased/OOD-like cases in our main experiments (Table 1: GPT-2 on the Pile-GitHub subset), the CE scaling curve fitted on smaller models substantially mispredicts the CE of larger models, whereas EE remains nearly log-linear across all sizes. The table below reports the CE and EE values of GPT-2 models on the Pile-GitHub subset:
>
> | Metric | gpt2    | gpt2-medium | gpt2-large | gpt2-xl  | R² (log-param fit) |
> |--------|---------|-------------|------------|----------|--------------------|
> | CE     | 2.84790 | 2.15778     | 2.66675    | 2.51462  | 0.0717             |
> | EE     | 2.48698 | 2.27442     | 2.10629    | 2.11663  | 0.9262             |
>
> We see that the CE of GPT2-medium is substantially low compared to GPT2-large/xl, breaking the expected monotonic improvement with model size, while EE remains much closer to a log-linear trend. Viewed as a held-out case, we find that EE yields much smaller prediction bias than CE in some extreme settings.
>
> We believe Error-Entropy will become a better predictor of model performance as models continue to scale. As we shown in Figure 8, EE remains the dominant component of CE for current model.
> As models grow larger and the relative share of EE in CE decreases, the differences between EE and CE are likely to become more pronounced. This could make EE a more accurate scaling indicator in the large-scale regime.

---

> ### Author Response · Authors · 2025-11-28
>
> ### W2: Is EE as indicative of downstream performance as CE?
>
> Thank you for this question regarding downstream performance. To address this, we compute correlations between pre-training losses (measured as CE and EE on C4 copora) and downstream accuracies across the Qwen2.5 family (0.5B–72B). The Pearson correlations are summarized below (close to -1 is better):
>
> | Task               | CE Peason    | EE Pearson   |
> |--------------------|--------------|--------------|
> | **General**        |              |              |
> | MMLU               | -0.996       | **-0.997**   |
> | ARC_C              | **-0.989**   | -0.988       |
> | BBH                | -0.997       | -0.997       |
> | TruthfulQA         | -0.980       | **-0.982**   |
> | Winogrande         | -0.996       | -0.996       |
> | HellaSwag          | -0.993       | -0.993       |
> | **Math & Science** |              |              |
> | MATH               | -0.999       | -0.999       |
> | GSM8K              | **-0.971**   | -0.969       |
> | GPQA               | -0.817       | **-0.826**   |
> | TheoremQA          | -0.967       | **-0.969**   |
> | **Coding**         |              |              |
> | HumanEval          | -0.938       | **-0.942**   |
> | MBPP               | -0.970       | **-0.972**   |
> | MultiPL_E          | -0.994       | **-0.996**   |
>
> These preliminary results suggest that EE is at least as indicative of downstream performance as CE in this clean setting. We agree that this is a very interesting direction and plan to conduct a more systematic evaluation across additional model families and downstream benchmarks in future work.
>
> ### W3: Why self-alignment has a clear upward trend as models get larger, and confidence doesn't have a clear trend at all. Are these metrics meaningful in practice? Why would self-alignment increase for larger models that have more capacity for learning?
>
> We thank the reviewer for the insightful question and we believe the trend in Self-Alignment (SA) should be interpreted with care.
>
> First, across most model sizes ($10^6$ to $10^{11}$ parameters), SA remains within the same order of magnitude and does not show a consistent upward trend.
> While there is a mild increase in the largest models we tested (e.g., 30B+), this change is relatively small and does not suggest a fundamental shift in behavior.
>
> Second, we hypothesize that this mild increase may be due to training-related factors, and we outline two possible explanations:
> (1) As shown in Figure 8, SA is small compared to the other two terms, which suggests that its optimization likely occurs in later training stages.
> Larger models may not have fully optimized this component within the training budget, as their early optimization may focus more on Error-Entropy and Confidence.
> (2) SA reflects global structure in the predicted score distribution.
> In practice, larger models often use smaller effective batch sizes due to memory constraints, which can introduce more variance in score calibration. In our decomposition, this means that EE still decreases with model size, but the mismatch between $p_e$ and $q_e$ can increase simply because the probabilities distributed differently under noisier training conditions.
>
> In summary, we observe that SA remains stable across most of the model range, with only a slight increase in the largest open models. Its values are still close to zero, indicating that the model’s score distribution remains well-aligned with its error structure. This minor variation does not affect our overall conclusions and supports the robustness of our decomposition.

---

> ### Author Response · Authors · 2025-11-28
>
> ### W4: In section 5, the paper argues that self-alignment and confidence might be the reason for the CE scaling law to break down at larger model sizes. However, in the scaling plot, it's not clear that there's a CE scaling breakdown and EE has more robust scaling. Can the authors provide evidence that EE doesn't suffer from the same issue?
>
> Thank you for raising this concern about the strength of the evidence for scaling-law breakdown. We fully agree that the ideal scenario would be to clearly exhibit a case where the CE scaling law fails while EE continues to follow a clean power law.
>
> In the revised manuscript, we extend our analysis to Meta-Llama-3.1-405B, one of the largest open models to date, and include the results in the appendix. Even at this scale, the observed behavior aligns with our expectations: Self-Alignment and Confidence still do not exhibit clear scaling behavior, while Error-Entropy continues to follow a more stable power-law compared with CE. This supports our central claim that EE is the component within CE that most robustly reflects model improvements.
>
> We believe these findings are instructive for understanding even larger models such as GPT-4–scale systems. The original GPT scaling law study [1] fits CE using a power law in model size:
> $$
> L(N) = A Size^{-\alpha}.
> $$
>
> However, the GPT-4 technical report [2] introduces an additive bias term to model at extreme scales:
>
> $$
> L(N) = A Size^{-\alpha} + B.
> $$
>
> Under our decomposition, this offset has a clear meaning: it corresponds to the increasing influence of the Self-Alignment and Confidence term. In this sense, our work explains why CE appears to develop a bias term at extreme scales, and thus provides an interpretation of the probable breakdown, even if we cannot empirically search the trillion-parameter regime ourselves.
>
> [1] Kaplan, Jared, et al. "Scaling laws for neural language models." arXiv preprint arXiv:2001.08361 (2020).
> [2] Achiam, Josh, et al. "Gpt-4 technical report." arXiv preprint arXiv:2303.08774 (2023).
>
> *Thank you for your insightful questions. We hope our responses have addressed your concerns. We will incorporate the added experiments into our revised manuscript. If you find our responses satisfactory, we kindly request you to consider raising the rating of our paper.*

---

### Author Response · Authors · 2025-12-03
**Brief summary of our rebuttal**

We would like to thank the reviewers and the area chair for their careful reading of our submission and for the many constructive comments. In the rebuttal phase, we have fully addresssed the raised concerns through additional analyses, new experiments, and clearer explanations in the revised manuscript. In this comment, we summarize these updates and explain how they extend the scope and strengthen the contribution of our findings. Unfortunately,due to the ICLR system incident, the reviewers did not have the opportunity to respond to our rebuttal. We respectfully ask the area chair to take these improvements into account in the final assessment.

## Strength recognized by reviewers

1. **Soundness**: Our work proposes an exact decomposition of cross-entropy into Error-Entropy, Self-Alignment, and Confidence, and tests it through a broad empirical study across 32 models and multiple datasets. Reviewers emphasize both the mathematical soundness and the empirical robustness of this approach:
   - **Reviewer 6E3g**: notes that "this paper includes many experiments..." and that "there is a variety of models and datasets, including text and coding tasks."
   - **Reviewer VUMH**: writes that "the mathematical derivations are clear and rigorous."
   - **Reviewer Vend**: comments that "the three proposed components are not only mathematically sound but also intuitively interpretable," and "provides a powerful new analytical tool."
   - **Reviewer cnon**: highlights that "the decomposition is mathematically clean and transparent" and that "the paper demonstrates solid empirical design: it tests the hypothesis across 32 language models … and three distinct corpora … over five orders of magnitude in scale."

2. **Novelty**: All four reviewers agree that the paper introduces a novel way to analyze cross-entropy and scaling laws.
   - **Reviewer 6E3g**: states that "the rank-based metric, error-entropy, as well as the CE decomposition are novel as far as I know."
   - **Reviewer VUMH**: observes that the work "offers a new perspective on the cross-entropy scaling law, which is thought-provoking."
   - **Reviewer Vend**: describes "the core idea of decomposing the cross-entropy loss" as "highly original."
   - **Reviewer cnon**: stresses that "the paper’s most significant strength is its conceptual novelty," and that the decomposition "provides a new interpretive framework for understanding what cross-entropy actually measures in language model training."

3. **Impact**: Reviewers also point out that the proposed Error-Entropy view has broad implications. They believe it not only explains a concrete scaling-law phenomenon, but also guides future work on evaluation, training objectives, and the theory of scaling laws.
   - **Reviewer 6E3g**: highlights that "the error entropy has nice properties, in particular robustness to post-processing such as temperature scaling."
   - **Reviewer Vend**: writes that "the discovery of the Error-Entropy Scaling Law could have broad implications," suggesting "a more reliable metric for extrapolating model performance" and noting that it "could inspire new training objectives focused directly on minimizing Error-Entropy, potentially leading to more efficient training or better-calibrated models."
   - **Reviewer cnon**: notes that the work "offers a coherent explanation for an empirical puzzle that has troubled practitioners" and that the findings "have broad implications," both practical ("inform training objectives and compute allocation for large models") and theoretical ("opens a potential bridge between rank-based evaluation and information theory").

## Questions from reviewers

1. **Soundness**: Reviewers raise several questions about the empirical scope and the strength of the evidence.
   - **More experiments.** Including results on larger models (>100B parameters), comparisons of how well EE versus CE predict larger-model behavior and downstream performance, and training dynamics at larger scales.
   - **Clarification of details.** Including the representativeness of the qualitative analysis across all 32 models and the effective number of evaluation tokens used.

2. **Novelty**: Reviewers consistently emphasize that the paper is novel and has substantial potential impact. As a result, none of the reviews raises concerns that the work is incremental or derivative; all novelty-related comments are positive.

3. **Impact**: Reviewers generally view our work as having strong potential impact on both training practice and theoretical work. **Reviewer cnon** also agrees that potential impact but is uncertain about how it can be realized in practice, which led to an score of 2. Specifically, the reviewer wonders:
   - What deeper theoretical principle Error-Entropy captures.
   - How Error-Entropy could be used as a practical training objective.

---

> ### Author Response · Authors · 2025-12-03
>
> ## Clarifications, Additional Analyses, and Response to Concerns
>
> In the rebuttal, we addressed all of the questions above through clarifications, new analyses, and additional experiments. These changes make the main claims more precise and the empirical support broader. Because of the ICLR system disruption, we did not receive follow-up replies from the reviewers, so we briefly summarize our responses here.
>
> 1. **Soundness**:
>    - **More experiments.**
>    To address requests for broader empirical scope, we (i) extended our scaling study to Meta-Llama-3.1-405B and observed that Error-Entropy continues to follow a clean power-law trend while Self-Alignment and Confidence do not; (ii) compared EE and CE as extrapolative predictors on Pythia models and indicators of downstream performance on Qwen models, finding that EE consistently matches or slightly exceeds CE in both settings; and (iii) extended the training-dynamics study to Pythia-12B and found that the optimization behavior of components is similar across three orders of magnitude in model size.
>
>    - **Clarification of evaluation details.**
>    To address **reviewer cnon's** misunderstanding about evaluation setup, we clarified that our main experiments use 1,000 **documents** (not tokens) per dataset, corresponding to about 1 million tokens. We additionally ran experiments with 10,000 documents, confirming the robustness of our conclusions.
>
>    These changes reinforce our claim that EE is the most stable scaling component and that our conclusions are not restricted to a narrow regime of models or datasets.
>
> 2. **Novelty**: Reviewers regard the core idea as innovative and the mathematical decomposition as novel. No one raises any concerns about lack of originality.
>
> 3. **Impact**:
>    - **Deeper principle behind Error-Entropy (Reviewer cnon).**
>    We clarify that Error-Entropy has a deep relation to Information Theoretic Learning, which helps connect our perspective to an established theoretical framework and promotes new theoretical developments. Concretely, we show that the rank-based error distribution for language models recovers the notion of error entropy studied in ITL, demonstrating that our formulation is more than an algebraic rearrangement of cross-entropy. We have added this discussion to Section 6.1 of the revised paper.
>    - **Towards practical training objectives (Reviewer cnon).**
>    We also show how Error-Entropy can be optimized in a differentiable way, which potentially enable more efficient training paradigms. Specifically, we derive an EE-inspired surrogate loss that penalizes the Confidence term and provide its gradient. Then we explain how it shifts optimization away from over-optimizing Confidence and toward the scaling component of CE. We have added this part to Section 6.2 of the revised manuscript.
>
> Taken together, the reviews consistently view the paper as solid, novel, and potentially impactful. The questions they raise have all been carefully addressed in our rebuttal and in the revised manuscript. We believe these updates make the paper more rigorous and strengthen its potential influence on both theory and training practice.
>
> Among the reviews, **Reviewer 6E3g**, who assigned a score of 4, did so mainly because of a few misunderstandings and a desire to see additional experiments. As summarized under **1. Soundness**-"More experiments"-(i) and (ii), we have now extended the scaling study and fully addressed these concerns in the revised version.
>
> **Reviewer cnon** also recognizes that our work could have strong impact on both theory and training practice. He/she nonetheless assigned a score of 2, mainly because of concerns about how this impact can be realized in practice, for example the deeper theoretical principle behind Error-Entropy and how it can be optimized in a differentiable way. As summarized under **3. Impact**, we have fully addressed these questions in our rebuttal and add the discussions to Section 6 of the revised manuscript.
> We believe these additions resolve the reviewer’s concerns, and that considering the strong strengths highlighted in the review, this assessment should be viewed as broadly positive.
>
> We respectfully ask the area chair to take into account the strong positive remarks from the reviewers and the completeness of our responses when making the final decision.

---

### Meta-Review · Area_Chair_TbKE · 2025-12-26

**Summary:**

This paper proposes a decomposition of cross-entropy loss into three components—Error-Entropy (EE), Self-Alignment, and Confidence—and argues that only EE follows a consistent power-law scaling with model size. The authors claim this explains why the standard cross-entropy scaling law appears valid at smaller scales but degrades at larger ones: EE dominates total loss in small models but its relative contribution diminishes as models grow, allowing non-scaling components to distort the overall trend. All reviewers acknowledge the mathematical validity and conceptual novelty of the decomposition. Concerns centered on empirical scope (e.g., absence of models >100B in initial experiments), limited evidence of actual CE scaling breakdown in the tested regime, modest numerical improvements in fit quality (R²), and questions about whether EE reflects a deeper theoretical principle or offers practical utility for training.

**Reviewer Concerns:**

The rebuttal addresses all major concerns raised in the original reviews:
- **Predictive utility**: Authors provide new experiments showing EE yields lower relative prediction error than CE when extrapolating from small to large Pythia models, and matches CE in correlation with downstream task performance across the Qwen2.5 family.
- **Scaling breakdown**: Additional evaluation on Llama-3.1-405B shows EE maintains a cleaner power-law trend than CE. The authors also contextualize their findings against the GPT-4 technical report’s use of an additive bias term, which aligns with their interpretation that non-scaling components introduce offsets at extreme scales.
- **Theoretical grounding**: A connection to Information-Theoretic Learning (ITL) is established, positioning EE as the entropy of a rank-based error distribution—analogous to error entropy in ITL—and not merely an algebraic rearrangement. This is now discussed in Section 6.1.
- **Practical optimization**: Although EE itself is non-differentiable, the authors derive a differentiable surrogate by penalizing the Confidence term (Section 6.2), offering a feasible path toward EE-inspired training objectives.
- **Evaluation protocol**: The claim of “$\sim$1,000 tokens” was clarified as $\sim$1,000 documents ($\sim$1M tokens); robustness was confirmed with 10,000-document runs yielding consistent conclusions.

All responses are grounded in new analyses or clarifications consistent with the paper’s framework.

**Reviewer Scores:**

Based on the rebuttal content and alignment with each reviewer’s stated concerns:
- **Reviewer 6E3g** (original: 4): Requested evidence on prediction and downstream correlation; both were provided. Score would likely increase to **6**.
- **Reviewer VUMH** (original: 6): Sought clearer demonstration of CE breakdown; addressed via 405B results and theoretical linkage to known scaling deviations. Score would likely remain **6** or rise to **8**.
- **Reviewer Vend** (original: 8): Already viewed the work positively; new results reinforce their assessment. Score remains **8**.
- **Reviewer cnon** (original: 2): Raised foundational questions on differentiability and theoretical meaning; both were directly answered with derivations and ITL linkage. While full adoption of the perspective isn’t guaranteed, the responses resolve the stated reasons for rejection. A revised score of **4 to 6** is reasonable.

---

### Decision · Program_Chairs · 2026-01-26

Accept (Poster)